# Inductive Bias and Spectral Properties of Single-Head Attention in High Dimensions

## Abstract

We study empirical risk minimization in a single-head tied-attention layer trained on synthetic high-dimensional sequence tasks, given by the recently introduced attention-indexed model. Using tools from random matrix theory, spin-glass physics, and approximate message passing, we derive sharp asymptotics for training and test errors, locate interpolation and recovery thresholds, and characterize the limiting spectral distribution of the learned weights. Weight decay induces an implicit nuclear-norm regularization, favoring low-rank query and key matrices. Leveraging this, we compare the standard factorized training of query and key matrices with a direct parameterization in which their product is trained element-wise, revealing the inductive bias introduced by the factorized form. Remarkably, the predicted spectral distribution echoes empirical trends reported in large-scale transformers, offering a theoretical perspective consistent with these phenomena.

## 1 Introduction and related work

Modern machine learning relies increasingly on attention mechanisms (Vaswani et al., 2017), which form the backbone of state-of-the-art models in natural language processing, vision, and beyond (Devlin et al., 2019; Dosovitskiy et al., 2020; Brown et al., 2020). Despite their empirical success, many questions related to understanding learning with attention layers remain open. For instance, the weight matrices of trained attention layers display striking and reproducible spectral patterns. Empirical studies report that the singular-value distributions of the query and key projections are far from random: they show a structured bulk with non-trivial tails (Martin et al., 2021; Staats et al., 2024), consistent with earlier observations of heavy-tailed and compressible spectra across deep networks (Mahoney & Martin, 2019). These regularities persist across architectures and training scales, yet their theoretical origin and implications for generalization remain largely unexplained. Why do such spectra emerge in attention weights, and what do they reveal about the inductive biases of the model?

Several empirical studies have further noted that attention projections are approximately low-rank. (Staats et al., 2024) found rapidly decaying spectra in pretrained transformers, while (Si et al., 2025) reported similar profiles in LLaMA models, showing that the low-rank tendency is present already at pretraining and persists through fine-tuning. Broader surveys confirm stable low-rank structure during training (Yunis et al., 2024), and parameter-efficient fine-tuning methods such as LoRA (Hu et al., 2022) exploit the fact that very low ranks suffice to capture relevant structure. Other works such have investigated the implicit regularization induced by optimization in attention networks (Tarzanagh et al., 2023; Sheen et al., 2024) and the induced low-rank bias caused by an explicit $\ell_2$ regularization on the weights (Kobayashi et al., 2024). Recent work further leverages spectral profiles for efficient adaptation (Si et al., 2025). While these findings are empirical, recent theoretical progress has linked weight decay on the query and key matrices to an effective nuclear-norm regularization of their product (Kobayashi et al., 2024), thereby formalizing a low-rank inductive bias in attention. This connects directly to earlier analyses of factorized parameterizations in feedforward networks, where weight decay implicitly promotes low-rank solutions through nuclear-norm penalties (Gunasekar et al., 2017; 2018; Arora et al., 2019; Galanti et al., 2024).

Despite these advances, we still lack a comprehensive theory connecting the low-rank implicit bias of the query and key matrices to generalization performance, and predicting the full spectral distribution of the learned weights. It also remains unclear how the standard factorized parameterization of query and key matrices compares to a direct, non-factorized parameterization of their product, or

whether attention continues to generalize in regimes where the size of the key and query projections is unrestricted.

A natural strategy for theoretical progress is to analyze simplified models in high-dimensional regimes, where the blessing of dimensionality (Donoho et al., 2000) often yields tractable characterizations of learning. This approach has driven major advances in the theory of two-layer networks via Gaussian single- and multi-index models (see, e.g., (Arous et al., 2021; Abbe et al., 2022; Ba et al., 2022; Arnaboldi et al., 2023; Damian et al., 2023; Bietti et al., 2025; Berthier et al., 2024)). In contrast, attention mechanisms remain far less explored. The few existing analyses (Cui et al., 2024; Troiani et al., 2025; Arnaboldi et al., 2025) focus on extremely low-rank regimes of order $\mathcal{O}(1)$, which cannot capture spectral properties observed in practice. More recently, attention-indexed models have been introduced (Boncoraglio et al., 2025), providing a principled data model for simplified attention layers. These models are especially appealing here, as they offer precisely the controlled setting in which the questions we raise—generalization, implicit low-rank bias, and spectral laws—can be addressed. While (Boncoraglio et al., 2025) focus on the Bayes-optimal estimator, our goal is to understand the performance of empirical risk minimization in attention layers. For this, we leverage recent progress on the asymptotic analysis of quadratic neural networks (Erba et al., 2025), which provides the technical tools to extend the Bayes-optimal analysis of (Boncoraglio et al., 2025) and characterize the solutions reached by empirical risk minimization.

In this paper, we carry out such an analysis for single-head tied attention trained by empirical risk minimization in the high-dimensional limit. We consider synthetic sequence-to-sequence and sequence-to-label tasks generated from the attention-indexed model (Boncoraglio et al., 2025), and study the effect of weight decay on the learned weight matrices. Using random matrix theory, spin-glass methods, and approximate message passing, we derive exact asymptotics for training and test errors, identify interpolation and recovery thresholds, and characterize the limiting singular-value distribution of the learned weights. Crucially, we exploit the connection between weight decay and nuclear-norm regularization to compare the inductive biases of the standard factorized parameterization of queries and keys with a direct non-factorized alternative. The different generalization abilities of these two variants have been investigated in (Zhang et al., 2025) in a in-context learning setting in the case of linear attention. We, instead, quantify and discuss the role of factorized and unfactorized weights in non-linear attention.

In this way, our theory not only predicts how generalization is preserved in overparameterized settings, but also reproduces the low-rank structure of attention weights with separated outliers, as empirically observed in transformers (Martin et al., 2021; Staats et al., 2024).

Methodologically, our analysis builds on tools from random matrix theory (Voiculescu et al., 1992; Dubova et al., 2023), recent results on Gaussian universality (Maillard & Kunisky, 2024) and extensive-rank analysis developed for ellipsoid-fitting (Maillard & Kunisky, 2024; Bandeira & Maillard, 2025), spin-glass methods (Talagrand, 2003; Thrampoulidis et al., 2020; Vilucchio et al., 2025), and in particular approximate message passing (Bayati & Montanari, 2011; Javanmard & Montanari, 2013; Berthier et al., 2020; Gerbelot & Berthier, 2023; Erba et al., 2025; Çakmak et al., 2024) to derive sharp asymptotics.

## 2 SETTING AND CONTRIBUTIONS

**Task and architecture.** We will consider a sequence-to-sequence (seq2seq) and a sequence-to-label (seq2lab) supervised learning task. Our analyses will apply to both these cases, actually showing that they are asymptotically equivalent. In the seq2seq task we consider a dataset $\mathcal{D} = \{\mathbf{x}_{\text{in}}^{\mu}, \mathbf{x}_{\text{out}}^{\mu}\}_{\mu=1}^{n}$ where both the input and output data $\mathbf{x}_{\text{in}}^{\mu}, \mathbf{x}_{\text{out}}^{\mu} \in \mathbb{R}^{T \times d}$ are sequences of $T$ tokens, each given through an embedding vector $\mathbf{x}_{\text{in},a}^{\mu} \in \mathbb{R}^{d}$ for $1 \leq a \leq T$. In the seq2lab task instead we consider a dataset $\mathcal{D} = \{\mathbf{x}_{\text{in}}^{\mu}, y^{\mu}\}_{\mu=1}^{n}$ where the output data $y^{\mu} \in \mathbb{R}^{T \times T}$ are matrices of $T \times T$ pair-wise tokens comparisons. In both cases, we aim at learning the input-output relationship through a parametrized function $\hat{f}(\mathbf{x}_{\text{in}}; W)$ of the form (respectively for the two tasks)

$$\begin{cases} \hat{f}_{\text{sq}}(\mathbf{x}_{\text{in}}; W) = A_W(\mathbf{x}_{\text{in}})\mathbf{x}_{\text{in}} \\ \hat{f}_{\text{lb}}(\mathbf{x}_{\text{in}}; W) = A_W(\mathbf{x}_{\text{in}}) \end{cases} \quad \text{with } A_W(\mathbf{x}_{\text{in}}) = \sigma_{\beta}\left( \frac{\mathbf{x}_{\text{in}} \frac{WW^T}{\sqrt{dm}} \mathbf{x}_{\text{in}}^T - \mathbb{E}_{\text{tr}}[\mathbf{x} \frac{WW^T}{\sqrt{dm}} \mathbf{x}^T]}{\sqrt{d}} \right) , \quad (1)$$

with weights $W \in \mathbb{R}^{d \times m}$, where $\sigma_{\beta}$ is the row-wise softmax activation at inverse temperature $\beta > 0$, and $A_W(\mathbf{x})$ is the attention matrix with tied key and query matrices. The factor $\sqrt{dm}$ is included to

ensure the matrix $S = WW^\top / \sqrt{md}$ has a spectrum of order $\mathcal{O}(1)$ in the high dimensional limit; similarly, the further factor $\sqrt{d}$ is included to ensure the whole attention inputs are of order $\mathcal{O}(1)$ in the same limit. Thus, in the seq2seq task, $\hat{f}$ is a tied attention layer with identity value matrix. In the seq2lab case instead, $\hat{f}$ outputs directly the attention matrix. We restrict our analysis to tied attention for analytical simplicity, but we remark that tied attention is expressive enough to showcase interesting phenomena, see for e.g. (Cui et al., 2024), and we do not expect a considerably different phenomenology to arise in this model in the untied case. Finally, $\mathbb{E}_{\mathrm{tr}}$ is the empirical average over $\mathbf{x}$ in the training set, and the corresponding term plays the role of a batch-centering, ensuring a mean-zero input to the activation.

We learn $W$ by *empirical risk minimization* (ERM) of the square loss with $\ell_2$ regularization (or equivalently, weight decay) that is commonly used in practice in LLMs. Respectively for the two tasks, it reads

$$\hat{W} = \arg\min \mathcal{L}(W), \quad \text{where} \quad \begin{cases} \mathcal{L}_{\mathrm{sq}}(W) := \frac{1}{d} \sum_{\mu=1}^n ||\mathbf{x}_{\mathrm{out}}^\mu - \hat{f}_{\mathrm{sq}}(\mathbf{x}_{\mathrm{in}}^\mu; W)||_2^F + \lambda \|W\|_F^2 \\ \mathcal{L}_{\mathrm{lb}}(W) := \sum_{\mu=1}^n ||y^\mu - \hat{f}_{\mathrm{lb}}(\mathbf{x}_{\mathrm{in}}^\mu; W)||_2^F + \lambda \|W\|_F^2 \end{cases} . \quad (2)$$

We measure the performance of the learned function $\hat{f}$ through the test errors

$$e_{\mathrm{test}}(\hat{f}) = \frac{1}{d}\mathbb{E}_{\mathbf{x}_{\mathrm{in}},\mathbf{x}_{\mathrm{out}}}||\mathbf{x}_{\mathrm{out}} - \hat{f}(\mathbf{x}_{\mathrm{in}})||_F^2, \quad e_{\mathrm{test}}(\hat{f}) = \mathbb{E}_{\mathbf{x}_{\mathrm{in}},y}||y - \hat{f}(\mathbf{x}_{\mathrm{in}})||_F^2, \quad (3)$$

where $\mathbb{E}$ stands for an average over an appropriate test set. We remark that our analytical framework can be extended to a larger class of losses and regularization, see Section 3.

**Input data model.** To allow for analytical tractability, we assume that the input sequences $\mathbf{x}_{\mathrm{in}}^\mu$ are Gaussian, i.e. that each token is independently given by $\mathbf{x}_{\mathrm{in},a}^\mu \sim \mathcal{N}(0, \mathbb{I}_d)$ for $1 \le a \le T$. In this case, it can be shown that the centering term in equation 1 concentrates to $\mathbb{I}_T \operatorname{Tr}(WW^T)$ in the large sample size regime $n \gg 1$. We remark that the high-dimensional setting we will consider allows for some universality way beyond the Gaussian hypothesis, in the same spirit as in (Xu et al., 2025, Assumption 2.2) and as proven by a number of recent works, e.g. (Montanari & Saeed, 2022; Dudeja et al., 2023; Dandi et al., 2023)

**Target function model.** Following the successes of the single-index models, and inspired by the recently introduced attention-indexed model (Boncoraglio et al., 2025), we consider a target function that lies within the expressivity class of the architecture in equation 1, and restrict the class of (possibly noisy) target functions to the ones of the form (respectively for the seq2seq and seq2lab tasks)

$$\mathbf{x}_{\mathrm{out}}^\mu = \sigma_{\beta_0}\left(R(\mathbf{x}_{\mathrm{in}})\right) \mathbf{x}_{\mathrm{in}} \quad \text{and} \quad y^\mu = \sigma_{\beta_0}\left(R(\mathbf{x}_{\mathrm{in}})\right) \quad (4)$$

where $R(\mathbf{x}_{\mathrm{in}}) \in \mathbb{R}^{T \times T}$ is a centered pre-activation matrix, and $\beta_0$ a softmax temperature possibly different from the learner's one $\beta$. We consider this model as we want to focus on the learnability of the token-to-token correlations exhibited by the output sequences. Concretely, we choose:

$$R(\mathbf{x}_{\mathrm{in}}) = \frac{\mathbf{x}_{\mathrm{in}} S_0 \mathbf{x}_{\mathrm{in}}^T - \mathbb{I}_T \operatorname{Tr}(S_0)}{\sqrt{d}} + \sqrt{\frac{\Delta}{2\mathbb{U}_T - \mathbb{I}_T}} \xi^\mu, \quad (5)$$

where $S_0$ is the target function weight matrix of rank $m_0$. We denoted $\xi^\mu \in \mathbb{R}^{T \times T}$ to be a symmetric standard Gaussian noise $\xi_{ab} = \xi_{ba} \sim N(0, 1)$ for all $1 \le a \le b \le T$, $\mathbb{U}_T$ is the $T \times T$ matrix of all ones, and the normalization of the variance ensures that the signal-to-noise ratio is uniform across tokens. The scalar $\Delta > 0$ plays the role of an inverse signal-to-noise ratio, encoding how much of the data structure is not captured by the attention-indexed component of the target. The intuition behind equation 5 is that the learning model is structurally limited to expressing models of attention that are bilinear in the input sequences: for this reason, we model all higher-order dependencies as (Gaussian) noise $\xi$. For the test set, we will consider samples distributed with the same distribution as the training set.

**High-dimensional limit.** A final ingredient to turn this setting into a model analyzable in a closed-form is the high-dimensional limit where the token embedding dimension $d \to \infty$, the number

of samples $n$ scales quadratically with $d$, and the rank of the target function weights $m$ diverges proportionally to $d$, i.e. the joint limit $d, n, m \to +\infty$ with

$$\alpha = n/d^2 = \mathcal{O}(1), \quad \kappa = m/d = \mathcal{O}(1), \quad \kappa_0 = m_0/d = \mathcal{O}(1). \qquad (6)$$

The quadratic sample complexity stems from the fact that, on the one hand, we seek a regime where the performance will change from trivially bad to perfect learning and, on the other hand, the number of degrees of freedom needed to determine the matrix $S_0$ is quadratic. In particular, for $\alpha \to 0$ (i.e. $n \ll d^2$) the ERM is the null estimator, i.e. the minimizer of the regularization term, while on the other hand for $\alpha \to +\infty$ (i.e. $n \gg d^2$) we have perfect recovery of the target weights. We finally stress that the following results do not necessarily need $m$ to scale linearly with the dimensions, but hold for any $m \geq d$.

We will further assume that in the same limit the empirical spectral density of $S_0$ (which may be either random or deterministic) converges to a limiting distribution $\mu_0$ with a finite first and second moment. When studying the limit of small target rank $\kappa_0 \to 0$, we will assume that the density of strictly positive eigenvalues of $S_0$ is bounded away from zero, and has bounded support. We note that the length of the sequence $T = \mathcal{O}(1)$ is considered finite in our work in line with other works on sequence multi-index or attention-indexed models (Cui et al., 2024; Troiani et al., 2025; Boncoraglio et al., 2025).

**Our main contributions.**

1. **Sharp learning curves and thresholds.** We present the first high-dimensional analysis of empirical risk minimization in single-head tied attention trained on synthetic tasks generated from the attention-indexed model (Boncoraglio et al., 2025) (in the extensive rank regime). Our results include exact formulas for training and test errors, and precise locations of interpolation and recovery thresholds, valid for sufficiently large $\kappa$, as quantified in Claim 1. We stress that that our results link global properties of the optimization dynamics to generalization, as well as clarifying the interplay between generalization and the spectral properties of the learned weights. Numerical experiments further show that, despite the non-convexity of the objective, gradient-based algorithms (Adam (Kingma & Ba, 2015), in particular) consistently reach the global minimum.

2. **Inductive bias from weight decay.** Building on the fact that $\ell_2$ regularization on $W$ induces an effective nuclear-norm penalty on $S = W^\top W/\sqrt{md}$ (Kobayashi et al., 2024), we quantify the resulting low-rank bias by computing the rank of the learned weight matrices in the high-dimensional limit (see Figure 1). This analysis explains, among other things, why over-parameterization (i.e., $m > \mathrm{rank}(S_0)$) does not harm generalization.

3. **Spectral law of the learned weights.** We derive the limiting singular-value distribution of the learned weights at the empirical loss optimum. As the sample complexity increases, the spectrum exhibits distinct regimes (Figure 1), with a bulk of outliers separating, in agreement with empirical observations (Martin & Mahoney, 2021; Martin et al., 2021; Staats et al., 2024).

4. **Factorized vs. non-factorized parameterization.** We compare the standard factorized training of query and key matrices (tied in our case) with direct element-wise training of their product, isolating the inductive bias introduced by factorization. This amounts to contrasting nuclear-norm versus Frobenius-norm regularization on the effective matrix $S = W^\top W/\sqrt{md}$. Although the non-factorized parameterization is strictly more expressive, its associated Frobenius regularization leads to significantly worse generalization (see Figure 3).

Methodologically, we map the estimation problem for $W$ in equation 2 to a generalized matrix sensing problem for the matrix $S = W^\top W/\sqrt{md} \in \mathbb{R}^{d \times d}$, where the $\ell_2$ regularization on $W$ translates into a nuclear norm regularization on $S$, imposing sparsity on the spectrum of the learned weights. We then study this equivalent matrix model by tools based on approximate message passing (Berthier et al., 2020; Gerbelot & Berthier, 2023) and their relation to convex and non-convex optimization (Loureiro et al., 2021; Vilucchio et al., 2025).

Beyond attention, our framework applies to a broader class of models with bilinear input–output structure. While our setting makes strong simplifications – single-head tied attention, Gaussian synthetic inputs, and fixed sequence length – these assumptions are deliberate. They isolate the statistical phenomena of interest (generalization, low-rank bias, and spectral structure) while keeping the analysis tractable. In this sense, the model plays the same role as Gaussian multi-index models

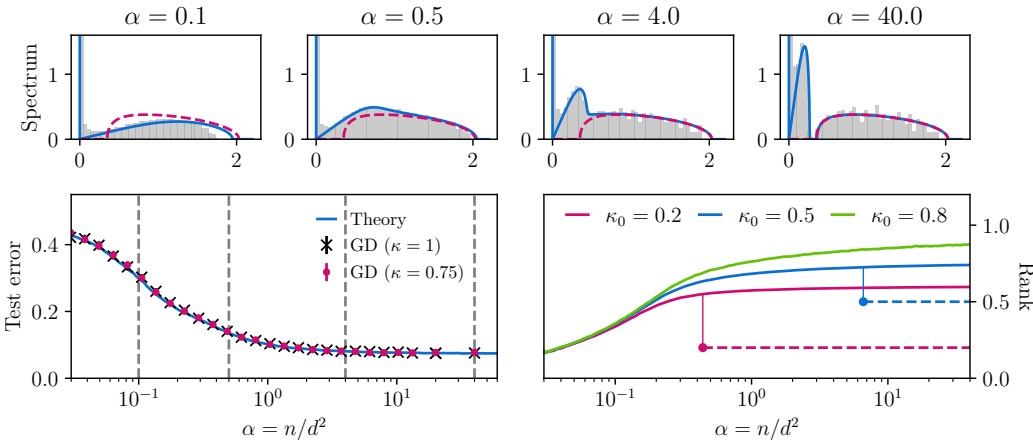

Figure 1: (Bottom left) Test error of the ERM estimator equation 2 (Claim 1, Eq. equation 12) compared with Adam simulations at $d = 100$ (64 instances, error bars = s.d.) as a function of sample size $\alpha = n/d^2$, where we use $\kappa = 0.75, 1$ (model rank) and parameters $\lambda = 0.01$, $\Delta = 0.5$, $T = 2$, $\beta = \beta_0 = 1$ for the MP target (Section 4, $\kappa_0 = 0.5$). Theory and simulations agree quantitatively over a broad range of $\alpha$. (Top) Singular value spectrum of the ERM estimator from theory equation 13 (blue) vs. Adam simulations (gray histograms) at $\alpha = 0.05, 0.5, 4, 40$ ($d = 200$, 64 runs, 2000 samples in the test set). The asymptotic spectrum of the target is shown in red dashed. The theory also captures the delta peak at zero. Agreement is good, up to finite-$d$ effects at small $\alpha$. For large $\alpha$ the spectrum splits into two bulks, with no impact on test error. (Bottom right) Predicted rank of the ERM estimator (solid) vs. $\alpha$ for $\kappa_0 = 0.2, 0.5, 0.8$. The right-most bulk mass (dashed) matches $\kappa_0$ once the bulk split occurs.

for two-layer networks: simplified yet revealing, and a starting point for theories of more realistic architectures.

## 3 MAIN TECHNICAL CLAIM

Our main technical result is the characterization of the properties of the global minima of the empirical loss of equation 2 in the high dimensional limit in terms of training and test error.

**Claim 1** (Asymptotics of ERM equation 2). *Consider the setting of Section 2. Define $\tilde{\lambda} = \sqrt{\kappa}\lambda$ and $\mu_\delta = \mu_0 \boxplus \mu_{\text{s.c.},\delta}$, where $\boxplus$ is the free convolution and $\mu_{\text{s.c.},\delta} = \sqrt{4\delta^2 - x^2}/(2\pi\delta^2)$ the semicircle distribution of radius $2\delta$ for $\delta > 0$, and $F_\delta$ its c.d.f.. Define also $\tilde{\sigma}_\beta(A) = \sigma_\beta(\sqrt{\mathbb{U}_T + \mathbb{I}_T}\,R)$ for any matrix $R \in \mathbb{R}^{T \times T}$. Call $(\Sigma^*, m^*, q^*, \hat{\Sigma}^*, \hat{m}^*, \hat{q}^*)$ the global minimizer of*

$$\Phi(\Sigma, m, q, \hat{\Sigma}, \hat{m}, \hat{q}) = \frac{\hat{q}\Sigma + 2\hat{m}m - \hat{\Sigma}q}{4} + \frac{n}{d^2}\mathcal{M}(\Sigma, m, q) - \frac{\hat{m}^2}{4\hat{\Sigma}}J\left(\frac{\sqrt{\hat{q}}}{\hat{m}}, \frac{2\tilde{\lambda}}{\hat{m}}\right), \quad (7)$$

*where*

$$\mathcal{M}(\Sigma, m, q) = \mathbb{E}_{z_0, z} \inf_{h \in \mathcal{S}(T)} \left\{ \frac{1}{2\Sigma} \sum_{\substack{a,b=1 \\ a \leq b}}^{T} (h_{ab} - z_{ab})^2 + \sum_{a,b=1}^{T} \left[\tilde{\sigma}_{\beta_0}(z_0)_{ab} - \tilde{\sigma}_\beta(h)_{ab}\right]^2 \right\}, \quad (8)$$

$$J(\delta, \epsilon) = \int_\epsilon^{+\infty} dx\, \mu_\delta(x)\,(x - \epsilon)^2,$$

*$\mathcal{S}(T)$ is the set of $T \times T$ symmetric matrices, and where $\mathbb{E}_{z_0, z}$ denotes the average over $\{z_{ab}, z_{0,ab}\}_{1 \leq a \leq b \leq T}$, which are Gaussian random variables*

$$\begin{bmatrix} z_{0,ab} \\ z_{ab} \end{bmatrix} \sim \mathcal{N}\left(\begin{bmatrix} 0 \\ 0 \end{bmatrix}; \begin{bmatrix} Q_0 + \Delta/2 & m \\ m & q \end{bmatrix}\right), \quad (9)$$

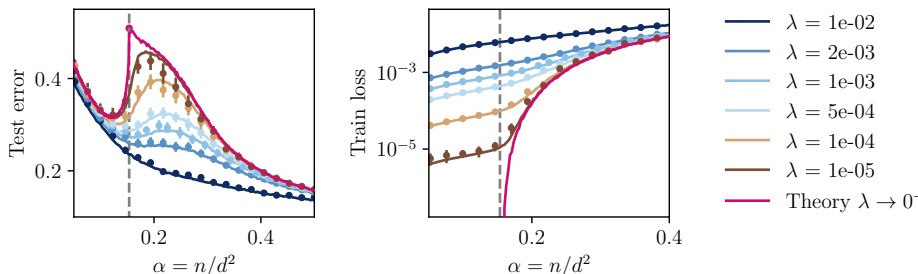

Figure 2: Test error (left) and training loss (right) as functions of the number of samples for decreasing regularization $\lambda$ ($\Delta = 0.5$, $T = 2$, $\beta = \beta_0 = 1$, $\kappa_0 = 0.5$, $\kappa = 1$). Solid lines: theoretical predictions; dots: Adam simulations at $d = 100$, averaged over 16 runs with 2000 samples in the test set. The gray dashed line marks the analytical interpolation threshold (Corollary 1), i.e., the largest $\alpha$ for which the ERM estimator fits the training data as $\lambda \to 0^+$. As expected, the training loss vanishes before interpolation (vertical log scale), while the test error exhibits a non-symmetric interpolation peak, distinct from the usual cusp-like shape.

*independently for each pair $a, b$. Assume that at the global minimum the replicon condition (Vilucchio et al., 2025) is satisfied:*

$$2\alpha \mathbb{E}_{z_0,z} \sum_{\substack{a,b,c,d=1 \\ a \leq b, \, c \leq d}}^{T} \left( \frac{\partial_{z_{ab}} p_{cd} - \delta_{ac}\delta_{bd}}{\Sigma} \right)^2 \int dx\, dy\, \mu_{\sqrt{\tilde{q}}/\hat{m}}(x)\, \mu_{\sqrt{\tilde{q}}/\hat{m}}(y)\, \frac{(\xi(x) - \xi(y))^2}{\hat{\Sigma}^2 (x-y)^2} < 1 \,,$$

(10)

*where $\xi(x) = ReLU(x - 2\tilde{\lambda}/\hat{m})$ and*

$$p(z_0, z, \Sigma) = \underset{h \in \mathcal{S}(T)}{\arg\inf} \left\{ \frac{1}{2\Sigma} \sum_{\substack{a,b=1 \\ a \leq b}}^{T} (h_{ab} - z_{ab})^2 + \sum_{a,b=1}^{T} [\tilde{\sigma}_{\beta_0}(z_0)_{ab} - \tilde{\sigma}_\beta(h)_{ab}]^2 \right\} .$$

(11)

*Then, for all values of $\alpha, \lambda > 0$, $\Delta \geq 0$ and $\kappa \geq 1 - F_{\sqrt{\tilde{q}}/\hat{m}}(2\tilde{\lambda}/\hat{m})$ any global minimum $\hat{W}$ of equation 2 satisfies*

$$\lim_{d \to \infty} \mathbb{E} e_{\text{test}}(\hat{W}) = \mathbb{E}_{z_0,z} \sum_{a,b=1}^{T} [\tilde{\sigma}_{\beta_0}(z_0)_{ab} - \tilde{\sigma}_\beta(z)_{ab}]^2 \,,$$

$$\lim_{d \to \infty} d^{-2} \mathbb{E}\mathcal{L}(\hat{W}) = \Phi(\Sigma, m, q, \hat{\Sigma}, \hat{m}, \hat{q})$$

(12)

*where all order parameters $(\Sigma, m, q, \hat{\Sigma}, \hat{m}, \hat{q})$ are evaluated at the global minimum defined above. Moreover, if $\mu_0$ has compact support, then the empirical singular value density of $\hat{W}/\sqrt[4]{md}$ satisfies*

$$\lim_{d \to \infty} \mathbb{E} \frac{1}{d} \sum_{i=1}^{d} \delta(x - \rho_i) = F_{\sqrt{\tilde{q}}/\hat{m}}(2\tilde{\lambda}/\hat{m})\delta(x) + I(x > 0) \left[ 2x\mu_{\sqrt{\tilde{q}}/\hat{m}}(x^2 + 2\tilde{\lambda}/\hat{m}) \right]$$

(13)

*where $\{\rho_i\}_{i=1}^{d}$ are the singular values of $\hat{W}$ and $I$ is the indicator function. In particular the rank of $\hat{W}$ is given by $1 - F_{\sqrt{\tilde{q}}/\hat{m}}(2\tilde{\lambda}/\hat{m})$.*

*Sketch of proof.* We state the main result as a claim rather than a theorem, since we only provide a proof sketch; completing all details would require substantial additional work. The missing steps, however, rely on classical arguments from the literature on high-dimensional inference. The proof outline is as follows, with further details in Appendix B.

The core of the approach is a reduction of a regularized version of the minimization in equation 2 to the one of a positive semi-definite (PSD) generalized matrix estimation with loss for the seq2seq task defined in the previous section

$$\tilde{\tilde{\mathcal{L}}}(S) := \frac{1}{d} \sum_{\mu=1}^{n} ||\tilde{\sigma}_{\beta_0}(S_0 X(\mathbf{x}^\mu))\mathbf{x}^\mu - \tilde{\sigma}_\beta(SX(\mathbf{x}^\mu))\mathbf{x}^\mu||_F^2 + \sqrt{md}\,\lambda \operatorname{Tr}(S)$$

(14)

where we defined $X_{ab}(\mathbf{x}) = (\mathbf{x}_a\mathbf{x}_b^T + \mathbf{x}_b\mathbf{x}_a^T - 2\delta_{ab}\mathbb{I}_d)/\sqrt{2d(\mathbb{U}_T + \mathbb{I}_T)}$, $S = W^TW/\sqrt{md}$ and $\tilde{\sigma}_\beta(R) = \sigma_\beta(\sqrt{\mathbb{U}_T + \mathbb{I}_T}\,R)$. We stress that here minimization is restricted to PSD matrices $S \succeq 0$. Then, one shows that the problem is asymptotically equivalent to the seq2lab task

$$\tilde{\mathcal{L}}(S) := \sum_{\mu=1}^n ||\tilde{\sigma}_{\beta_0}(S_0 X(\mathbf{x}^\mu)) - \tilde{\sigma}_\beta(S X(\mathbf{x}^\mu))||_F^2 + \sqrt{md}\,\lambda\,\mathrm{Tr}(S)\,, \tag{15}$$

by a law of large numbers argument applied to the loss, as done for e.g. in (Cui et al., 2024). Then, we use the Gaussian universality principle, that allows to replace each matrix $X_{ab}(\mathbf{x}^\mu)$ by a random independent Wigner matrix $G_{ab}^\mu$, following closely the steps of (Maillard et al., 2024; Xu et al., 2025; Erba et al., 2025). This achieves a reduction of the original problem to a rank-penalized generalized matrix recovery problem with $\mathrm{GOE}(d)$ sensing matrices.

It remains to characterize the high-dimensional asymptotics of this non-convex problem, a key difference from related convex settings such as (Erba et al., 2025). To this end, we build directly on the framework of (Vilucchio et al., 2025), combining the Gaussian Min–Max Theorem (GMT) (Stojnic, 2009; Thrampoulidis et al., 2014), which provides sharp lower bounds, with Approximate Message Passing (AMP) (Donoho et al., 2009), which achieves these bounds algorithmically under the replicon condition. Our contribution is to extend this methodology to the attention setting: we construct a suitable AMP with a non-separable prior, relying on the detailed state-evolution analysis of (Berthier et al., 2020; Gerbelot & Berthier, 2023). Moreover, since the standard Gordon bound applies to regression, we rely here on its classification analogue developed in (Thrampoulidis et al., 2020). Taken together, these extensions show that AMP is tight in our setting, yielding a full asymptotic characterization of the global minimizer of equation 15. $\qquad\square$

Claim 1 provides an asymptotic prediction for train and test error, as well as a characterization of the singular values of the optimal attention weights. We also remark that our results do not depend on the network width $m$, as long as $m \geq \mathrm{rank}(\hat{W}\hat{W}^T)$ (and the r.h.s. is characterized analytically in Claim 1), meaning that massive over-parameterization is also included in our framework, incurring no penalty in terms of test error and explaining why rank-constrained architectures can still generalize well. We also stress that our framework can be adapted to more general losses and regularizations. Specifically, we can treat any loss and any data model that is asymptotically of the form $\sum_{\mu=1}^n \ell\left(S_0 X(\mathbf{x}^\mu); SX(\mathbf{x}^\mu)\right)$ (i.e. depending bi-linearly on the data) with any regularization inducing a rotationally-invariant penalty on the effective matrix $S = WW^\top/\sqrt{md}$. This includes, for e.g., models with mismatched nonlinearities between data and learner, classification tasks, and more. We provide the equations for the more general case in Appendix B.2.1.

Note that Claim 1 requires evaluating the replicon condition 10, which guarantees convergence of the AMP algorithm (Bolthausen, 2014). Although a general proof is difficult and direct evaluation can be numerically demanding, in our experiments AMP did converge (the replicon condition being a convergence criterion for AMP), and gradient-based algorithms reached the same fixed point. This provides strong empirical evidence that the replicon condition is satisfied in our setting.

## 4 CONSEQUENCES OF THE MAIN CLAIM

To illustrate our results, in all the figures we will focus on the *Marchenko-Pastur (MP) target* case, where the weights $S_0$ are such that the limiting distribution satisfies $\mu_0(x) = \sqrt{\kappa_0}\mu_{\mathrm{M.P.}}(\sqrt{\kappa_0}x)$, where $\mu_{\mathrm{M.P.}}$ is the Marchenko-Pastur distribution (Marchenko & Pastur, 1967) with parameter $0 < \kappa_0 < 1$, and $Q_0 = 1 + \kappa_0$. For example, this is the case in which the target attention matrix has the form $S_0 = W_0 W_0^T/\sqrt{m_0 d}$, where $W_0 \in \mathbb{R}^{d\times m_0}$ has i.i.d. components extracted from a distribution with zero mean and unit variance, and $\kappa_0 = m_0/d = \mathrm{rank}(S_0)/d$. We believe that the overall phenomenology we showcase here is valid qualitatively for generic $S_0$ with $\mathrm{rank}(S_0) = \kappa_0/d$ and positive eigenvalues bounded away from the origin. Additionally, we consider the case of matched softmax temperatures $\beta = \beta_0 = 1$ (noticing that a mismatch in temperatures can always be reabsorbed in a rescaling of the regularization). Appendix E provides some discussion of the dependence on the temperature and the number of tokens.

**Sharp learning curves at quadratic sample complexity.** In Figure 1 and Figure 2 we plot the test error predicted by our theory as a function of the sample ratio $\alpha = n/d^2$ for $\Delta = 0.5$. We observe a monotone decreasing behavior at large values of regularization $\lambda$ (Figure 1), while at lower values of $\lambda$ a characteristic interpolation peak (with oddly asymmetric shape) appears around the maximal

number of samples that can be perfectly fit by the model architecture, the so-called interpolation threshold (Figure 2). Claim 1 provides an analytic prediction for the interpolation threshold, i.e. the value $\alpha_{\text{interp}}$ before which the ERM estimator for $\lambda \to 0^+$ achieves zero training loss. In the noiseless case $\Delta = 0$, we can also predict analytically the perfect recovery threshold, i.e. the value $\alpha_{\text{perfect}}$ after which the ERM estimator achieves perfect generalization.

**Corollary 1** (Interpolation and perfect recovery thresholds). *Consider the setting of Section 2 with $\lambda = 0$, $\Delta \geq 0$ and $T \geq 2$. Then, there exists a value of sample ratio $\alpha_{\text{interp}}$ such that the training loss at its global minimum is zero for $\alpha < \alpha_{\text{interp}}$ (perfect fit of the training set), and strictly positive for $\alpha > \alpha_{\text{interp}}$, and $\alpha_{\text{interp}}$ satisfies*

$$\alpha_{\text{interp}} = \frac{\partial_1 J\left(\bar{\delta}, 0\right)}{2\bar{\delta}(T^2 + T - 2)} + o_d(1) \quad \text{where} \quad Q_0 + \frac{\Delta}{2} = J\left(\bar{\delta}, 0\right) - \frac{\bar{\delta}}{2}\partial_1 J\left(\bar{\delta}, 0\right) , \quad (16)$$

*with $J$ defined in equation 8. Moreover, if $\Delta = 0$, there exists a value of sample ratio $\alpha_{\text{perfect}}$ such that the test error at the global minimum is zero for $\alpha > \alpha_{\text{perfect}}$ (perfect generalization), and strictly positive for $\alpha < \alpha_{\text{perfect}}$, determined as follows. Call $\bar{c}$ the solution of the equation*

$$M_{\text{s.c.}}^{(1)}(c) - cM_{\text{s.c.}}^{(0)}(c) + \frac{c}{1 - \kappa_0} = 0 \quad \text{where} \quad M_{\text{s.c.}}^{(k)}(x) = \int_{-2}^{x} dx \, \mu_{\text{s.c.}}(x) \, x^k \quad (17)$$

*for $0 < \kappa_0 < 1$. Then,*

$$\alpha_{\text{strong}} = \frac{1}{T^2 + T - 2}\left[1 - (1 - \kappa_0)^2\left(M_{\text{s.c.}}^{(2)}(c) - cM_{\text{s.c.}}^{(1)}(c)\right)\right] + o_d(1) . \quad (18)$$

Corollary 1 follows from a mapping of the theory of softmax attention to that of linear attention that holds for $\alpha \leq \alpha_{\text{interp}}$ and $\lambda \to 0^+$, allowing to adapt results from (Erba et al., 2025). We discuss the mapping in Appendix C, and we stress that for $\alpha > \alpha_{\text{interp}}$ or $\lambda$ bounded away from zero the mapping breaks down, leading to genuinely different behavior. We also remark that the same mapping held for the Bayes-optimal estimator (Boncoraglio et al., 2025).

**Behavior of gradient descent.** A natural question is whether gradient-based methods, routinely used to train neural networks, can reach global minima of the non-convex loss in equation 2, or whether they remain stuck in spurious local minima. In Figure 1 and Figure 2 we compare our theory with numerical results of runs of Adam (Kingma & Ba, 2015) (in the standard PyTorch implementation (PyTorch Core Team, 2025)) run on the loss of equation 2 with $d = 100$ and averaging over 32 different instances, and $m = d$. We observe an excellent match for both the test error and the training loss, which given the non-convexity of the loss is a highly non-trivial phenomenon, whose analytical characterization is a challenging problem that we leave for future work. Additionally, in Figure 1 we plot Adam experiments with same parameters, but $\kappa = m/d = 3/4$. We observe that even with this additional constraint, Adam finds the global minimum. The question of asking when gradient-based methods reach global minima in attention models is, indeed, crucial and many works such as (Vasudeva et al., 2024; Huang et al., 2024; Li et al., 2024) have investigated its optimization guarantees have in various settings. Our contribution is complementary. The analysis of the algorithmic dynamics in our high-dimensional setting is left for future work, as we believe it to require non-trivial technical advancements.

**Inductive bias: weight decay implies nuclear norm on attention.** The $\ell_2$ regularization over the weights $W$ naturally translates to a nuclear norm regularization in the equivalent generalized matrix problem, see (Kobayashi et al., 2024) and Appendix A, naturally favoring model weights configurations with an effective lower width (i.e., implementable with fewer hidden units): a weight decay in $W$ thus implies a low-rank learned attention matrix. This partly explains why learning with low-rank weights (usually done for computational reasons) does not negatively affect generalization despite reducing expressivity. In Figure 1 (bottom right) we quantify this low-rank bias by computing the rank of the learned weight matrices in the high-dimensional limit (see Figure 1 and equation 13).

In the limit $\lambda \to 0^+$, thanks to the mapping to linear attention (Appendix C), we can study analytically the limit $\kappa_0 \to 0^+$ for the factorized model. In this limit, the inductive bias is particularly beneficial. Among the analytical results that we inherit from (Erba et al., 2025), we remark that the estimation error decays to zero as $\mathcal{O}(\Delta dm_0/n)$, meaning that learning is efficiently achieved on a sample scale $\mathcal{O}(dm_0) \ll \mathcal{O}(d^2)$, and the rank of the learned weights is of order $\mathcal{O}(\kappa_0^{3/5})$.

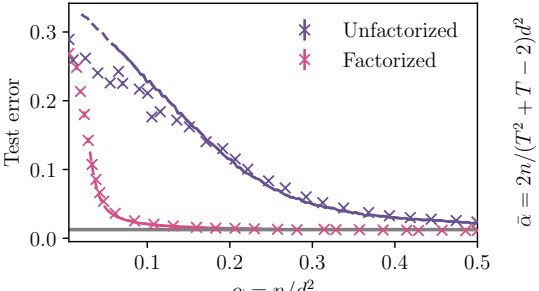 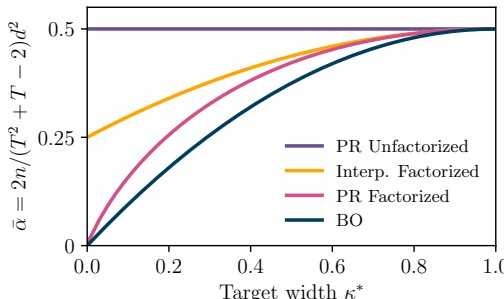

Figure 3: (Left) Comparison between the standard factorized key and query training equation 2 and training the non-factorized matrix equation 20, at optimal regularization (cross-validated on the test error), with $\kappa_0 = 0.05$, $\Delta = 0.05$, $T = 2$, $\beta = \beta_0 = 1$. Solid lines are theory (Claim 1, and its adaptation to non-factorized case Appendix D), while dots are numerical simulations with Adam at $d = 100$ averaged over 8 instances with 2000 samples in the test set. The optimal regularization is the one with smallest test error in $10^{-6}$ and $10^{-1}$. We see that the standard factorized model achieves much lower test error at all values of $\alpha$. (Right) Analytical thresholds for vanishing regularization in the noiseless case $\Delta = 0$. We plot the Bayes-optimal perfect recovery from (Boncoraglio et al., 2025) (lower-bound to the perfect recovery of any estimator), the perfect recovery of the attention model equation 2 from Corollary 1, the interpolation threshold from Corollary 1, and the perfect recovery of the non-factorized attention model equation 20.

**Exact spectral law of the learned weights.** Claim 1, equation 13 provides an asymptotic prediction for the limiting density of singular values of the learned weights. The spectral density depends on the problem's parameters only through two scalars $\delta = \sqrt{\hat{q}}/\hat{m}$ and $\epsilon = 2\tilde{\lambda}/\hat{m}$, and is given by a noisy, cut-off version of the target's spectral density, with noise $\delta$ strength roughly correlating with the test error (larger noise, larger test error), and cutoff $\epsilon$ soft-thresholding smaller eigenvalues to zero (implementing the low-rank-inducing nuclear norm regularization). This form of the spectrum is quite simple, allowing to gain intuition and classify the possible behaviors of the spectrum easily. For example, for the MP target case, we first observe a single bulk phase at low number of samples, splitting into two bulks as the number of samples increases (Figure 1). In the two-bulks phase, we have a bulk of larger eigenvalues correlating with the target spectrum and with the target mass $\kappa_0$, while the second bulk gathers smaller eigenvalues for which the regularization was not strong enough to soft-threshold to zero. Figure 1 compares the prediction for the spectral density with the spectrum of the weights learned in experiments with Adam at dimension $d = 200$, finding an excellent agreement. We also observe the inductive bias in action: the learned spectra feature a possibly large fraction of zero singular values at all values of $\alpha$.

The spectra in Figure 2 can be compared with experimentally observed ones (Martin et al., 2021; Martin & Mahoney, 2021; Staats et al., 2024). Our model with MP target distribution reproduces one of the main features observed in real architectures: the appearance of outliers (in our case, an entire bulk of outliers) as the performance of the model improves. The other main feature, i.e. power-law tails, are not observed in the MP target. We conjecture that a model with heavy-tailed target distribution would feature such phenomenology, but leave such exploration for future work.

**Comparison with non-factorized parameterization.** Given the form of the target function in equation 4 and equation 5, a natural baseline to consider is learning a non-factorized attention $S$ through the empirical risk minimization of the model (for e.g. in the seq2seq task)

$$\hat{f}_{\text{L2}}(\mathbf{x}_{\text{in}}; S) = \sigma_\beta \left( \frac{\mathbf{x}_{\text{in}} S \mathbf{x}_{\text{in}}^T - \mathbb{E}_{\text{tr}} \text{Tr}(\mathbf{x}_{\text{in}} S \mathbf{x}_{\text{in}}^T)}{\sqrt{d}} \right) \mathbf{x}_{\text{in}} , \tag{19}$$

where we impose that $S$ is symmetric (but neither PSD nor factorized), and we train on the loss

$$\mathcal{L}_{\text{L2}}(S) := \frac{1}{d} \sum_{\mu=1}^{n} ||\mathbf{x}_{\text{out}}^\mu - \hat{f}_{\text{L2}}(\mathbf{x}_{\text{in}}^\mu; S)||_2^F + \lambda ||S||_F^2 . \tag{20}$$

In other words, we perform generalized ridge regression on a model matched to the data-generating process. The Frobenius-regularized estimator underperforms the nuclear-norm–regularized one whenever the target is sufficiently low-rank. In the noisy case, we compare the test error of the factorized ERM estimator of equation 2 with the non-factorized Frobenius estimator of equation 20 at optimal regularization (cross-validated, Fig. 3 left). The factorized model consistently outperforms, fully exploiting the induced low-rank bias. In the extreme low-rank regime $m_0 \ll d$, it achieves vanishing error at sample scale $\mathcal{O}(dm_0)$, whereas the non-factorized model requires $\mathcal{O}(d^2)$.

In the noiseless case and in the limit $\lambda \to 0^+$, we compare recovery thresholds, i.e. the value of $\alpha = n/d^2$ beyond which the test error vanishes. In Fig. 3 (right) we show these thresholds together with the interpolation threshold (Corollary 1) and the Bayes-optimal strong recovery threshold (Boncoraglio et al., 2025). The non-factorized model reaches perfect recovery only at $\alpha = 1/2$, while the factorized model does so much earlier for all $0 < \kappa_0 < 1$; as $\kappa_0 \to 0^+$, the gap scales as $\mathcal{O}(\kappa_0^{-1})$.

## 5 CONCLUSION AND LIMITATIONS

We presented a high-dimensional analysis of empirical risk minimization in single-head tied attention, deriving exact learning curves, thresholds, and spectral laws. Our results quantify the emergence of low-rank structure, and reveal the inductive difference between factorized and non-factorized parameterizations, and are in line with empirical spectra observed in transformers.

While our framework captures essential spectral and generalization phenomena, it has limitations. First, we restrict to tied single-head attention with identity values, whereas modern architectures involve multiple heads, untied projections, and residual pathways. Second, we assume Gaussian input embeddings with structured input-output relationship under supervised learning, leaving open the extension to natural structured inputs from which one learns in self-supervised manner.

Our results suggest several avenues for empirical investigation. In particular, they motivate systematic measurements of the spectral properties of transformer weights across training scales, to test the predicted low-rank bias and spectral regimes. Looking ahead, several theoretical follow-ups appear promising. Extending the framework to untied and multi-head attention would test the robustness of our predictions. Incorporating power-law tails into our spectral analysis may help formalize the empirical scaling laws reported across large transformer models. Finally, a deeper understanding of optimization dynamics, beyond our asymptotic characterization of the minima, remains an important open challenge.

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

# A    Inductive Nuclear Norm Bias for Tied and Untied weights

We briefly recall here the mathematical reasons behind the appearance of the *inductive nuclear norm*, following arguments that go back to classical variational characterizations of atomic norms and were first emphasized in the deep learning context by (Neyshabur et al., 2015; Gunasekar et al., 2017; Soudry et al., 2018; Pesme & Flammarion, 2023). These arguments are quite general: whenever a predictor is parameterized in a factorized form (for instance as the product of two matrices, or as a diagonal interaction of two vectors), an $\ell_2$-type penalty on the factors induces, via an AM–GM–style variational identity, an $\ell_1$ or nuclear norm penalty on the effective predictor. More recent works (Maillard & Kunisky, 2024; Xu et al., 2025) have also used these ideas.

**Case 1 (Tied weight):**    Let $W \in \mathbb{R}^{d \times p}$ and $M := W^\top W \in \mathbb{R}^{p \times p}$. Assume the loss depends on $W$ only via $M$, i.e. there exists $\Phi : \mathbb{S}_+^p \to \mathbb{R}$ such that

$$\mathcal{L}(W) = \Phi(W^\top W) + \lambda \|W\|_F^2.$$

Then the following problems are equivalent in optimal value, and their minimizers correspond via $M = W^\top W$:

$$\min_{W \in \mathbb{R}^{d \times p}} \Phi(W^\top W) + \lambda \|W\|_F^2 \quad \equiv \quad \min_{M \succeq 0, \ \mathrm{rank}(M) \leq d} \Phi(M) + \lambda \|M\|_*. \qquad (\star)$$

*Proof.* Since $M = W^\top W \succeq 0$ with $\mathrm{rank}(M) \leq d$, and if $s_i(W)$ are the singular values of $W$, then the eigenvalues of $M$ are $s_i(W)^2$. Hence

$$\|W\|_F^2 = \sum_i s_i(W)^2 = \sum_i \lambda_i(M) = \|M\|_*.$$

Because $\Phi$ depends only on $M$, we can replace $W$ by $M$ and obtain $(\star)$. Moreover, any feasible $M \succeq 0$ with $\mathrm{rank}(M) \leq d$ admits a factorization $M = W^\top W$; choosing such a $W$ makes the two objectives equal.    $\square$

**Case 2 (Untied weights):**    Let $U \in \mathbb{R}^{m \times r}$, $V \in \mathbb{R}^{n \times r}$, and $M := UV^\top \in \mathbb{R}^{m \times n}$. Assume the loss depends on $(U, V)$ only through the product $M$, i.e.

$$\mathcal{L}(U, V) \ = \ \Psi(UV^\top) \ + \ \frac{\lambda}{2}\big(\|U\|_F^2 + \|V\|_F^2\big),$$

for some (possibly nonconvex) function $\Psi : \mathbb{R}^{m \times n} \to \mathbb{R}$. We show the exact equivalence

$$\min_{U,V} \Psi(UV^\top) + \tfrac{\lambda}{2}(\|U\|_F^2 + \|V\|_F^2) \quad \equiv \quad \min_{M \in \mathbb{R}^{m \times n}} \Psi(M) + \lambda \|M\|_*. \qquad (\dagger)$$

**Matrix AM–GM analogue (variational identity for $\|\cdot\|_*$).**    The key inequality is the matrix analogue of AM–GM: for any factorization $M = UV^\top$,

$$\|M\|_* \ \leq \ \tfrac{1}{2}\big(\|U\|_F^2 + \|V\|_F^2\big), \qquad (A)$$

with equality attained by a specific SVD-based factorization.

*Proof of (A).* Use the dual characterization of the nuclear norm:

$$\|M\|_* \ = \ \max_{\|A\|_2 \leq 1} \langle M, A \rangle \ = \ \max_{\|A\|_2 \leq 1} \langle UV^\top, A \rangle \ = \ \max_{\|A\|_2 \leq 1} \langle U, AV \rangle,$$

where $\langle X, Y \rangle := \mathrm{Tr}(X^\top Y)$ and $\|\cdot\|_2$ is the spectral norm. For any $A$ with $\|A\|_2 \leq 1$, apply Cauchy–Schwarz and the scalar AM–GM (or $ab \leq \frac{1}{2}(a^2 + b^2)$):

$$\langle U, AV \rangle \ \leq \ \|U\|_F \|AV\|_F \ \leq \ \|U\|_F \|A\|_2 \|V\|_F \ \leq \ \|U\|_F \|V\|_F \ \leq \ \tfrac{1}{2}\big(\|U\|_F^2 + \|V\|_F^2\big).$$

Taking the maximum over $\|A\|_2 \leq 1$ yields (A).

**Tightness of (A).** Let the compact SVD of $M$ be $M = P\Sigma Q^\top$ with $\Sigma = \mathrm{diag}(\sigma_1, \ldots, \sigma_\rho)$, $\sigma_i \geq 0$. Choose the *balanced SVD factorization*

$$U := P\Sigma^{1/2}, \qquad V := Q\Sigma^{1/2}.$$

Then $M = UV^\top$ and

$$\|U\|_F^2 + \|V\|_F^2 = \mathrm{Tr}(\Sigma) + \mathrm{Tr}(\Sigma) = 2\sum_{i=1}^{\rho} \sigma_i = 2\|M\|_*.$$

Hence equality holds in (A):

$$\|M\|_* = \min_{M=UV^\top} \tfrac{1}{2}\big(\|U\|_F^2 + \|V\|_F^2\big), \tag{B}$$

and the minimum is attained by the balanced SVD factorization.

**Equivalence of objectives.** Because $\Psi$ depends on $(U, V)$ only through $M = UV^\top$, we may fix $M$ and minimize the regularizer over all factorizations $M = UV^\top$. By (B),

$$\inf_{U,V:\, UV^\top = M} \frac{\lambda}{2}\big(\|U\|_F^2 + \|V\|_F^2\big) = \lambda\|M\|_*.$$

Therefore

$$\inf_{U,V} \Psi(UV^\top) + \tfrac{\lambda}{2}(\|U\|_F^2 + \|V\|_F^2) = \inf_M \Psi(M) + \lambda\|M\|_*,$$

which is exactly (†).

Notice that no convexity of $\Psi$ is required for the equality of optimal values; convexity only affects algorithmic guarantees.

## B  DERIVATION OF CLAIM 1

In this paper, we provide an informal justification for Claim 1. Claim 1 can be proven by going through the following steps (already presented in the main text):

- Reduction to the PSD generalized matrix estimation problem

$$\tilde{\tilde{\mathcal{L}}}(S) := \frac{1}{d}\sum_{\mu=1}^{n} ||\tilde{\sigma}_\beta(SX(\mathbf{x}^\mu))\mathbf{x}^\mu - \tilde{\sigma}_{\beta_0}(S_0 X(\mathbf{x}^\mu))\mathbf{x}^\mu||_F^2 + \sqrt{md}\,\lambda\,\mathrm{Tr}(S) \tag{21}$$

  where $X_{ab}(\mathbf{x}) = (\mathbf{x}_a \mathbf{x}_b^T + \mathbf{x}_b \mathbf{x}_a^T - 2\delta_{ab}\mathbb{I}_d)/\sqrt{2d(\mathbb{U}_T + \mathbb{I}_T)}$, $S = W^T W/\sqrt{md} \succeq 0$ and $\tilde{\sigma} = (A) = \sigma(\sqrt{1 + \mathbb{I}}\,A)$. This step is discussed in Appendix A.

- Asymptotic equivalence of sequence-to-sequence and sequence-to-attention tasks. One needs to show (as done for e.g. in (Cui et al., 2024)) that equation 14 is asymptotically the same as equation 15

$$\tilde{\mathcal{L}}(S) := \sum_{\mu=1}^{n} ||\tilde{\sigma}_\beta(SX(\mathbf{x}^\mu)) - \tilde{\sigma}_{\beta_0}(S_0 X(\mathbf{x}^\mu))||_F^2 + d\tilde{\lambda}\,\mathrm{Tr}(S), \tag{22}$$

  where we called $\tilde{\lambda} = \sqrt{m/d}\lambda = \sqrt{\kappa}\lambda$. To do this, one would need to prove a law-of-large-numbers kind of argument of the kind

$$\frac{1}{d}||F(\mathbf{x})\mathbf{x}||_F^2 = \frac{1}{d}F(\mathbf{x})\mathbf{x}\mathbf{x}^T F(\mathbf{x})^T \approx F(\mathbf{x})F(\mathbf{x})^T \tag{23}$$

  where we used that $\mathbb{E}_\mathbf{x}\mathbf{x}\mathbf{x}^T/d = \mathbb{I}_T$.

- Gaussian universality, i.e. replacement of each matrix $X_{ab}(\mathbf{x}^\mu)$ by a random independent Wigner matrix $G_{ab}^\mu$. This step is *a priori* non-trivial, and its proof would require generalizing the arguments of (Maillard et al., 2024; Xu et al., 2025; Erba et al., 2025) to the case of multiple tokens (the case $T = 1$ is instead included in (Erba et al., 2025)). We do not foresee any technical roadblock here, as this step requires only promoting scalar outputs to finite-dimensional vectorial ones in all such proofs.

- Analysis of the final empirical risk minimization problem

$$\tilde{\mathcal{L}}(S) := \frac{1}{d} \sum_{\mu=1}^{n} ||\sigma_\beta(SG^\mu) - \sigma_{\beta_0}(S_0 G^\mu)||_F^2 + d\,\tilde{\lambda}\,\mathrm{Tr}(S)\,. \tag{24}$$

The asymptotic analysis of this loss can be done by adapting the analysis in (Erba et al., 2025) (corresponding to the single token case $T = 1$ with linear activation) to multiple tokens, in the same spirit as the generalization of single-index models to multiple tokens performed in (Cui et al., 2024). In practice this reduces to:

- Writing down an appropriate Approximate Message Passing (AMP) algorithm (which we present explicitly in the following) whose fixed point are local minima of the loss in equation 24.
- Using AMP theory to derive a set of low-dimensional state evolution equations that track AMP step-by-step, and use them to characterize the properties of its fixed points.
- Among the fixed points of state evolution, select the one with lowest value of the training loss (describing then the properties of global minima).

The only non-trivial point here is that this program is *a priori* that for non-convex losses such as equation 24 AMP may not converge signaling the onset of so-called replica symmetry breaking. Nonetheless, in (Vilucchio et al., 2025) the authors rigorously show that the described procedure is correct whenever the replicon condition in equation 10 is satisfied. Again one would need to generalize their proof (an AMP upper bound coupled with a lower-bound in the line of (Stojnic, 2009; Thrampoulidis et al., 2014; 2020)) to multiple tokens, but that should not pose any roadblock.

We devote the rest of the section to write down the suitable AMP algorithm used in the sketch of the proof, in the more generic setting of multi-token generalized matrix sensing.

## B.1 GENERIC SETTING

Consider the data model

$$y^\mu = g_0(\{\mathrm{Tr}(S_0 Z_a^\mu)\}_{a=1}^{L_{\mathrm{in}}})\,, \tag{25}$$

where $Z_a^\mu$ are GOE($d$) matrices, $S_0 \in \mathbb{R}^{d\times d}$ and $g_0 : \mathbb{R}^{L_{\mathrm{in}}} \to \mathbb{R}^{L_{\mathrm{out}}}$. $g_0$ can be a stochastic function, in which case we assume that its stochasticity is independent and identical for each sample $\mu$. Consider the empirical risk minimization problem over $d\times d$ symmetric matrices

$$\hat{S} = \underset{S\in\mathcal{C}}{\arg\min}\, L(S)\,, \qquad L(S) = \sum_\mu \ell\bigg(y^\mu; \{\mathrm{Tr}(S Z_{ab}^\mu)\}_{a=1}^{L_{\mathrm{in}}}\bigg) + R(S)\,, \tag{26}$$

for $\ell : \mathbb{R}^{L_{\mathrm{out}}} \times \mathbb{R}^{L_{\mathrm{in}}} \to \mathbb{R}$, $R : \mathbb{R}^{d\times d} \to \mathbb{R}$ a rotationally invariant regularization (i.e. $R(S) = R(OSO^T)$ for all $d$-dimensional rotation matrices $O$) and $\mathcal{C}$ is a rotationally-invariant subset of the set of $d\times d$ symmetric matrices.

**Special case I: main text model.** In the main text, we consider $L_{\mathrm{in}} = T(T+1)/2$ (interpreted as $T \times T$ symmetric matrices), $L_{\mathrm{out}} = T^2$, $g_0$ is the row-wise softmax with Gaussian noise in the input

$$g_0(A) = \sigma([A + \sqrt{\Delta}\xi]/\sqrt{2 - \mathbb{I}_T})\,, \tag{27}$$

where $\xi^\mu \in \mathbb{R}^{T\times T}$ is a symmetric standard Gaussian noise $\xi_{ab} = \xi_{ba} \sim N(0,1)$ for all $1 \le a \le b \le T$,

$$\ell(y, \hat{y}) = \sum_{a,b=1}^{T} (y_{ab} - \hat{y}_{ab})^2 \quad \text{and} \quad R(S) = d\tilde{\lambda}\,\mathrm{Tr}(S)\,. \tag{28}$$

Finally, $\mathcal{C}$ is the set of PSD symmetric $d\times d$ matrices.

**Special case II: main text model with linear attention.** In this case, we consider $L_{\text{in}} = T(T+1)/2$ and $L_{\text{out}} = T(T+1)/2$ (interpreted as $T \times T$ symmetric matrices), $g_0$ is

$$g_0(A) = \frac{A + \sqrt{\Delta}\xi}{\sqrt{2 - \mathbb{I}_T}} \,, \tag{29}$$

where $\xi^\mu \in \mathbb{R}^{T \times T}$ is a symmetric standard Gaussian noise $\xi_{ab} = \xi_{ba} \sim N(0,1)$ for all $1 \le a \le b \le T$, and

$$\ell(y, \hat{y}) = \sum_{a,b=1}^{T} (y_{ab} - \hat{y}_{ab})^2 \quad \text{and} \quad R(S) = d\tilde{\lambda} \operatorname{Tr}(S) \,. \tag{30}$$

Finally, $\mathcal{C}$ is the set of PSD symmetric $d \times d$ matrices. The case of single token $T = 1$ reduces to (Erba et al., 2025).

## B.2 Mapping to a vector-weights model with coupled regularization

We consider the mapping from vec : $\operatorname{Sym}_d \to \mathbb{R}^{d(d+1)/2}$ (which conveniently maps the Frobenius scalar product in $\operatorname{Sym}_d$ given by $\langle A\,; B \rangle = \operatorname{Tr}(AB)$ to the standard Euclidean scalar product in $\mathbb{R}^{d(d+1)/2}$) given by

$$\operatorname{vec}(A)_{(ab)} = \left\langle b^{(ab)}\,; A \right\rangle = \sqrt{2 - \delta_{ab}} A_{ab} \,, \tag{31}$$

under the choice of orthonormal basis

$$b_{ij}^{(aa)} = \delta_{ia}\delta_{ja} \,, \quad b_{ij}^{(ab)} = \frac{\delta_{ia}\delta_{jb} + \delta_{ib}\delta_{ja}}{\sqrt{2}} \,. \tag{32}$$

Here $(ab)$ stands for the ordered pair of $1 \le a \le b \le d$, and we denote $A_{ij}$ as the $i, j$ entry of a matrix $A$, while as $A_{(ab)}$ the component of matrix $A$ onto the basis element $b^{(ab)}$. Let us denote $d(d+1)/2 = D$ (we will often use $D \approx d^2/2$ as we will be interested in the leading order in $d$).

Our input data is given, for each sample $\mu$, by $L_{\text{in}}$ $d \times d$ symmetric matrices $\{Z_a\}_{a=1}^{L_{\text{in}}}$ that in the Gaussian equivalent model we treat as independent GOEs. Then the sensing vectors satisfy

$$A_{a,(ij)} = \sqrt{\frac{d}{2}} \operatorname{vec}(Z_a)_{(ij)} \sim N(0, \mathbb{I}_D) \tag{33}$$

for $1 \le a \le L_{\text{in}}$, and $(ij)$ means the dimension indices $1 \le i \le j \le d$. Moreover, if we define

$$w_{(ij)} = \sqrt{\frac{d}{2}} \operatorname{vec}(S)_{(ij)} \,, \tag{34}$$

we have that

$$\operatorname{Tr}(SZ_a) = \sum_{(ij)} Z_{a,(ij)} S_{(ij)} = \sqrt{\frac{2}{d}} \sum_{(ij)} A_{a,(ij)} S_{(ij)} = \frac{\sqrt{2}}{\sqrt{D}} \sum_{(ij)} A_{a,(ij)} w_{(ij)} \,,$$

$$\frac{1}{d} \operatorname{Tr}(S^2) = \frac{1}{d} \sum_{(ij)} S_{(ij)}^2 = \frac{2}{d^2} \sum_{(ij)} w_{(ij)}^2 = \frac{1}{D} \sum_{(ij)} w_{(ij)}^2 \,. \tag{35}$$

In this formulation, under Gaussian equivalence, we see that our model is just a sequence single-index model of the form (Cui et al., 2024) with coupled prior term.

Finally, the loss in the vectorial model reads

$$\tilde{\ell}(y = g_0(z_0), z) = \ell(y = g_0(\sqrt{2}z_0), \sqrt{2}z) \tag{36}$$

due to the factor $\sqrt{2}$ in equation 35.

### B.2.1 Approximate Message Passing

Under the mapping of Section B.2 we can directly combine the single token derivation of AMP given in (Erba et al., 2025) (which holds for generic single-token losses and generic rotational-invariant regularization), with multi-token AMP given in (Cui et al., 2024) to obtain the following AMP algorithm with fixed points given by local minima of the loss in equation 26. We stress that the multi-token extension can be mapped to the standard treatment of AMP, amounting to a non-separable non-linearity along the samples dimension. Hence the mapping from ERM to AMP of (Erba et al., 2025) can be adapted to the multi-token case.

**AMP in vector notation.** In vector notation, the labels are generated as

$$y = g_0(\{\sqrt{2}D^{-1/2}\sum_{(ij)} A^0_{\mu,a,(ij)}w_{(ij)}\}^{L_{in}}_{a=1}) \tag{37}$$

and the AMP reads (here $\mu = 1, \ldots, n$, $a = 1, \ldots, L_{in}$ and $1 \leq i \leq j \leq d$)

$$w^t_{(ij)} = \sqrt{\frac{d}{2}}\mathrm{vec}\left(\phi(\sqrt{\frac{2}{d}}\mathrm{mat}\left(\Gamma^{t-1}\right), \Lambda^{t-1})\right)_{(ij)}$$

$$V^t = \frac{1}{D}(\mathrm{div}\phi)(\sqrt{\frac{2}{d}}\mathrm{mat}\left(\Gamma^{t-1}\right), \Lambda^{t-1})$$

$$\omega^t_{\mu,a} = \frac{1}{\sqrt{D}}\sum_{(ij)} A_{\mu,a,(ij)}w^t_{(ij)} - \theta(t \geq 1)V^t f^{t-1}_{\mu,a}$$

$$f^t_{\mu,a} = \frac{\mathrm{prox}(y_\mu, \omega^t_\mu, V^t)_a - \omega^t_{\mu,a}}{V} \tag{38}$$

$$\Lambda^t = -\frac{1}{D}\sum_{a=1}^{L_{in}}\sum_{\mu=1}^{n}\partial_{\omega_{\mu,a}}f^t_{\mu,a}$$

$$\Gamma^t_{(ij)} = \frac{1}{\sqrt{D}}\sum_{a=1}^{L_{in}}\sum_{\mu=1}^{n}A_{\mu,a,(ij)}f^t_{\mu,a} + \Lambda^t w^t_{(ij)}$$

where

$$\mathrm{prox}(y, \omega, V) = \underset{h\in\mathbb{R}^{L_{in}}}{\arg\inf}\left\{\frac{1}{2V}\sum_{a=1}^{L_{in}}(h_a - \omega_a)^2 + \ell(y, \sqrt{2}h)\right\}$$

$$\phi(M = ODO^T, \Lambda) = O\underset{T\in\mathcal{C}}{\arg\min}\left[\frac{1}{d^2}R(T) + \frac{\Lambda}{4}\sum_{i=1}^{d}T_i^2 - \frac{1}{2}\sum_{i=1}^{d}T_i D_i\right]O^T \tag{39}$$

where $M$ is a $d$-dimensional symmetric matrix with eigen-decomposition $M = ODO^T$.

**AMP in matrix notation.** If we map back all quantities to their original matrix shape, we obtain the following equivalent AMP algorithm. The labels are generated as

$$y^\mu = g_0(\{\mathrm{Tr}(S_0 Z^\mu_a)\}^{L_{in}}_{a=1}) \tag{40}$$

and the AMP reads (here $\mu = 1, \ldots, n$, $a = 1, \ldots, L_{in}$ and $1 \leq i, j \leq d$)

$$S^t = \phi(\Theta^{t-1}, \Lambda^{t-1})$$

$$V^t = \frac{2}{d^2}(\mathrm{div}\phi)(\Theta^{t-1}, \Lambda^{t-1})$$

$$\Omega^t_{\mu,a} = \mathrm{Tr}(S^t Z^\mu_a) - \theta(t \geq 1)V^t F^{t-1}_{\mu,a}$$

$$F^t_{\mu,a} = \frac{\mathrm{prox}_{\mathrm{mat}}(y^\mu, \Omega^t_\mu, V^t)_a - \Omega^t_{\mu,a}}{V} \tag{41}$$

$$\Lambda^t = -\frac{2}{d^2}\sum_{a=1}^{L_{in}}\sum_{\mu=1}^{n}\partial_{\Omega_{\mu,a}}F^t_{\mu,a}$$

$$\Theta^t_{ij} = \frac{1}{d}\sum_{a=1}^{L_{in}}\sum_{\mu=1}^{n}Z^\mu_{a,ij}F^t_{\mu,a} + \Lambda^t S^t_{ij}$$

where $\phi$ is the same as in the vector case, while

$$\mathrm{prox}(y, \omega, V)_{\mathrm{mat}} = \underset{h\in\mathbb{R}^{L_{in}}}{\arg\inf}\left\{\frac{1}{4V}\sum_{a=1}^{L_{in}}(h_a - \omega_a)^2 + \ell(y, h)\right\}. \tag{42}$$

**State evolution.** The iterations of both algorithms can be tracked by the following state evolution equations.

$$
\begin{cases}
\hat{q}^t & = \frac{2n}{d^2(\Sigma^t)^2}\mathbb{E}_{z_0,z}\sum_{a=1}^{L_{\text{in}}}(p_a - z_a)^2 \\
\hat{\Sigma}^t & = \frac{2n}{d^2\Sigma^t}\left[L_{\text{in}} - \mathbb{E}_{z_0,z}\sum_{a=1}^{L_{\text{in}}}\frac{Q_0 z_a p_a - m(z_0)_a p_a}{Q_0 q - m^2}\right] \\
\hat{m}^t & = \frac{2n}{d^2\Sigma^t}\mathbb{E}_{z_0,z}\sum_{a=1}^{L_{\text{in}}}\frac{q(z_0)_a p_a - m z_a p_a}{Q_0 q - m^2} \\
m^{t+1} & = -\partial_{\hat{m}}\Psi(\hat{\Sigma}^t, \hat{q}^t, \hat{m}^t) \\
q^{t+1} & = 2\partial_{\hat{\Sigma}}\Psi(\hat{\Sigma}^t, \hat{q}^t, \hat{m}^t) \\
\Sigma^{t+1} & = -2\partial_{\hat{q}}\Psi(\hat{\Sigma}^t, \hat{q}^t, \hat{m}^t)
\end{cases}
\tag{43}
$$

where

$$
p_a = \operatorname*{arginf}_{h\in\mathbb{R}^{L_{\text{in}}}}\left\{\frac{1}{2\Sigma^t}\sum_{a=1}^{L_{\text{in}}}(h_a - \omega_a)^2 + \ell(g_0(\sqrt{2}\{z_0\}), \sqrt{2}h)\right\},
\tag{44}
$$

$$
\Psi(\hat{\Sigma}, \hat{q}, \hat{m}) = \frac{2}{d}\mathbb{E}_D \min_{T\in\mathcal{C}}\left[\frac{1}{d^2}R(T) + \frac{\hat{\Sigma}}{4}\sum_{i=1}^d T_i^2 - \frac{1}{2}\sum_{i=1}^d T_i D_i\right],
$$

where the average $\mathbb{E}_D$ is over the spectrum $D$ of the matrix $\hat{m}S_0 + \sqrt{\hat{q}}Z$ with $Z \sim \text{GOE}(d)$, while $\mathbb{E}_{z_0,z}$ is over two Gaussian vectors $z_0, z \in \mathbb{R}^{L_{\text{in}}}$ such that, independently for all components $a = 1, \ldots, L_{\text{in}}$

$$
\begin{bmatrix}(z_0)_a \\ z_a\end{bmatrix} \sim \mathcal{N}\left(\begin{bmatrix}0 \\ 0\end{bmatrix}; \begin{bmatrix}Q_0 & m^t \\ m^t & q^t\end{bmatrix}\right),
\tag{45}
$$

and over any stochasticity of the activation $g_0$.

**Observables.** Call $T^*$ is the optimizer in $\Psi$ of equation 44. Then, by state evolution, at the global minimum $\hat{W}$ of equation 26, we have that the spectral density of $\hat{W}\hat{W}^T/\sqrt{md}$ converges to the one of $T^*$. Moreover, pre-activations converge to Gaussians of the form

$$
\begin{bmatrix}\text{Tr}(S_0 Z_a) \\ \text{Tr}(S Z_a)\end{bmatrix} \sim \mathcal{N}\left(\begin{bmatrix}0 \\ 0\end{bmatrix}; \begin{bmatrix}2Q_0 & 2m \\ 2m & 2q\end{bmatrix}\right)
\tag{46}
$$

when evaluated on data $Z$ that is not included in the training set. Otherwise, the target's pre-activations are still Gaussian with zero mean and variance $2Q_0$, but the learned model's pre-activations are given by

$$
\text{Tr}(S Z_a) \sim \sqrt{2}p_a.
\tag{47}
$$

With that in mind, we have that the training error (without regularization term) at a fixed point of AMP is given by

$$
e_{\text{train}} = \mathbb{E}_{z_0,z}\sum_{a=1}^{L_{\text{in}}}\ell\left(g_0(\sqrt{2}z_0); \sqrt{2}p\right),
\tag{48}
$$

The regularization part of the loss can be computed as $d^{-2}R(T^*)$, and any test loss (intended as a comparison of the output of the target and learned function) can be instead computed as

$$
e_{\text{test}} = \mathbb{E}_{z_0,z}\sum_{a=1}^{L_{\text{in}}}\ell_{\text{test}}\left(g_0(\sqrt{2}z_0); \sqrt{2}z\right).
\tag{49}
$$

All averages are as defined for the state evolution equations.

**Variational formulation.** It can be checked (Cui et al., 2024) that equation 7 is stationary at the fixed point of equation 43, and that it matches the training loss.

**Replicon condition.** The replicon condition is the linear stability condition of AMP under perturbations of a fixed points. A derivation is given in (Erba et al., 2025). In our case, we have

$$
\frac{2\alpha}{\hat{\Sigma}^2\Sigma^2}\mathbb{E}_{z_0,z}\sum_{\substack{a,b,c,d=1 \\ a\leq b,\, c\leq d}}^T (\partial_{z_{ab}}p_{cd} - \delta_{ac}\delta_{bd})^2 \frac{2}{d^2}\mathbb{E}_D\sum_{\substack{1\leq i\leq j\leq d \\ 1\leq k\leq l\leq d}}\partial_{M_{ij}}\phi(M = ODO^T, \hat{\Sigma}) < 1
\tag{50}
$$

where the averages are the same as for state evolution.

### B.3 SIMPLIFICATIONS IN THE MAIN TEXT SETTING

In the main text, we consider $L_{\text{in}} = T(T+1)/2$ (interpreted as $T \times T$ symmetric matrices), $L_{\text{out}} = T^2$, $g_0$ is the row-wise softmax with Gaussian noise in the input

$$g_0(A) = \sigma_{\beta_0}([A + \sqrt{\Delta}\xi]/\sqrt{2\mathbb{U}_T - \mathbb{I}_T}), \tag{51}$$

where $\xi^\mu \in \mathbb{R}^{T \times T}$ is a symmetric standard Gaussian noise $\xi_{ab} = \xi_{ba} \sim N(0,1)$ for all $1 \le a \le b \le T$,

$$\ell(y,z) = \sum_{a,b=1}^{T} (y_{ab} - \sigma_\beta(z/\sqrt{2\mathbb{U}_T - \mathbb{I}_T})_{ab})^2 \quad \text{and} \quad R(S) = d\tilde{\lambda}\operatorname{Tr}(S). \tag{52}$$

In this case,

$$\phi(M = ODO^T, \Lambda) = O\operatorname{diag}\left(\frac{\operatorname{ReLU}(D_{ii} - 2\tilde{\lambda})}{\Lambda} \Big| i = 1, \ldots, d\right)O^T$$

$$\Psi(\hat{\Sigma}, \hat{q}, \hat{m}) = -\frac{\hat{m}^2}{2\hat{\Sigma}}J(\sqrt{\hat{q}}/\hat{m}, 2\tilde{\lambda}/\hat{m}) \tag{53}$$

$$J(a,b) = \int_b^{+\infty} dx\, \mu_a(x)\,(x-b)^2.$$

where $\mu_a = \mu_{\text{s.c.}} \boxplus \mu_0$. Notice that the noise can be treated simply by altering the second moment of $z_0$ in the state evolution equations from $Q_0$ to $Q_0 \to Q_0 + \Delta/2$ (where the factor 2 comes from gathering the $\sqrt{2}$ factor present in the state evolution equations).

**Observables.** The training and test errors are given by

$$e_{\text{train}} = \mathbb{E}_{z_0,z} \sum_{a=1}^{L_{\text{in}}} ||\tilde{\sigma}_{\beta_0}(\sqrt{2}z_0) - \tilde{\sigma}_\beta(\sqrt{2}p)||_F^2,$$

$$\tag{54}$$

$$e_{\text{test}} = \mathbb{E}_{z_0,z} \sum_{a=1}^{L_{\text{in}}} ||\tilde{\sigma}_{\beta_0}(\sqrt{2}z_0) - \tilde{\sigma}_\beta(\sqrt{2}z)||_F^2.$$

The spectral density of $\hat{W}\hat{W}^T/\sqrt{md}$ converges to that of $T^*$, which is a shifted and cropped version of the spectral density of $S_0 + \sqrt{\hat{q}}/\hat{m}Z$ for $Z \sim \operatorname{GOE}(d)$, giving the expression in the main text.

**State evolution.** This gives the following state equations (at the fixed point)

$$q = \frac{\hat{m}^2}{\hat{\Sigma}^2} J\left(\frac{\sqrt{\hat{q}}}{\hat{m}}, \frac{2\tilde{\lambda}}{\hat{m}}\right)$$

$$m = \frac{\hat{m}J\left(\frac{\sqrt{\hat{q}}}{\hat{m}}, \frac{2\tilde{\lambda}}{\hat{m}}\right) - \tilde{\lambda}\partial_2 J\left(\frac{\sqrt{\hat{q}}}{\hat{m}}, \frac{2\tilde{\lambda}}{\hat{m}}\right) - \frac{\sqrt{\hat{q}}}{2}\partial_1 J\left(\frac{\sqrt{\hat{q}}}{\hat{m}}, \frac{2\tilde{\lambda}}{\hat{m}}\right)}{\hat{\Sigma}}$$

$$\Sigma\hat{\Sigma} = \frac{\hat{m}}{2\sqrt{\hat{q}}}\partial_1 J\left(\frac{\sqrt{\hat{q}}}{\hat{m}}, \frac{2\tilde{\lambda}}{\hat{m}}\right) \tag{55}$$

$$\Sigma\hat{\Sigma} = 2\alpha L - 2\alpha\mathbb{E}_{z_0,z}\sum_{a=1}^{L_{\text{in}}} \frac{Q_0 z_a p_a - m z_{0,a} p_a}{Q_0 q - m^2}$$

$$\Sigma\hat{m} = 2\alpha\mathbb{E}_{z_0,z}\sum_{a=1}^{L_{\text{in}}} \frac{q z_{0,a} p_a - m z_a p_a}{Q_0 q - m^2}$$

$$\hat{q}\Sigma^2 = 2\alpha\mathbb{E}_{z_0,z}\sum_{a=1}^{L_{\text{in}}} (p_a - z_a)^2$$

where $\alpha = n/d^2$ and

$$p_a = \underset{h \in \mathbb{R}^{L_{\text{in}}}}{\operatorname{arginf}} \left\{ \frac{1}{2\Sigma} \sum_{a=1}^{L_{\text{in}}} (h_a - \omega_a)^2 + \sum_{a=1}^{L_{\text{out}}} (\tilde{\sigma}_{\beta_0}(\sqrt{2}z_0)_a - \tilde{\sigma}_\beta(\sqrt{2}h)_a)^2 \right\}. \tag{56}$$

For the replicon, the only thing that changes is that the regularization dependent part becomes the same as presented in (Erba et al., 2025), giving the expression of Claim 1.

**Small regularization limit $\lambda \to 0^+$.** To study the small regularization limit before interpolation (where a vanishing regularization would lead to degenerate global minima), one can perform the change of variables $\hat{m} \to \hat{m}\tilde{\lambda}$, $\hat{q} \to \hat{q}\tilde{\lambda}^2$, $\Sigma \to \Sigma/\tilde{\lambda}$ and $\hat{\Sigma} \to \hat{\Sigma}\tilde{\lambda}$ to obtain

$$q = \frac{\hat{m}^2}{\hat{\Sigma}^2} J\left(\frac{\sqrt{\hat{q}}}{\hat{m}}, \frac{2}{\hat{m}}\right)$$

$$m = \frac{\hat{m}J\left(\frac{\sqrt{\hat{q}}}{\hat{m}}, \frac{2}{\hat{m}}\right) - \tilde{\lambda}\partial_2 J\left(\frac{\sqrt{\hat{q}}}{\hat{m}}, \frac{2}{\hat{m}}\right) - \frac{\sqrt{\hat{q}}}{2}\partial_1 J\left(\frac{\sqrt{\hat{q}}}{\hat{m}}, \frac{2}{\hat{m}}\right)}{\hat{\Sigma}}$$

$$\Sigma\hat{\Sigma} = \frac{\hat{m}}{2\sqrt{\hat{q}}}\partial_1 J\left(\frac{\sqrt{\hat{q}}}{\hat{m}}, \frac{2}{\hat{m}}\right)$$

$$\Sigma\hat{\Sigma} = 2\alpha L - 2\alpha \mathbb{E}_{z_0,z} \sum_{a=1}^{L_{\text{in}}} \frac{Q_0 z_a p_a - m z_{0,a} p_a}{Q_0 q - m^2} \tag{57}$$

$$\Sigma\hat{m} = 2\alpha \mathbb{E}_{z_0,z} \sum_{a=1}^{L_{\text{in}}} \frac{q z_{0,a} p_a - m z_a p_a}{Q_0 q - m^2}$$

$$\hat{q}\Sigma^2 = 2\alpha \mathbb{E}_{z_0,z} \sum_{a=1}^{L_{\text{in}}} (p_a - z_a)^2$$

with

$$p_a = \underset{h \in \mathbb{R}^{L_{\text{in}}}}{\operatorname{arginf}} \left\{ \frac{\tilde{\lambda}}{2\Sigma} \sum_{a=1}^{L_{\text{in}}} (h_a - \omega_a)^2 + \sum_{a=1}^{L_{\text{out}}} (\tilde{\sigma}_{\beta_0}(\sqrt{2}z_0)_a - \tilde{\sigma}_\beta(\sqrt{2}h)_a)^2 . \right\} \tag{58}$$

We then see that for $\tilde{\lambda} \to 0^+$ the last expression reduces to

$$p_a = \underset{h \in \mathbb{R}^{L_{\text{in}}}}{\operatorname{arginf}} \left\{ \sum_{a=1}^{L_{\text{out}}} (\tilde{\sigma}_{\beta_0}(\sqrt{2}z_0)_a - \tilde{\sigma}_\beta(\sqrt{2}h)_a)^2 \right\}. \tag{59}$$

with the prescription that if such arginf is degenerate, one should pick the arginf closest to $z$ in L2 distance. This set of equations is valid as long as $\Sigma > 0$ and not diverging. At interpolation we expect $\Sigma \to 0$, as there the curvature of the loss (proportional to $\Sigma^{-1}$) diverges.

After interpolation, there is no need to change variable. One can put directly $\lambda \to 0^+$ in the original equation 43. This set of equations is valid as long as $\Sigma > 0$ and not diverging. At interpolation we expect $\Sigma \to +\infty$, as there the curvature of the loss (proportional to $\Sigma^{-1}$) goes to zero.

## C  REDUCTION TO LINEAR ATTENTION

Let us consider the setting of the main text in the $\lambda \to 0^+$ limit, before interpolation (i.e. when there exists multiple sets of weights that perfectly fit the training dataset). Then, we need to solve equation 66. We now show that here, for the case of softmax-softmax studied in the main text, the equations reduce to a rescaled version of the state equations for the case of linear single token attention (Erba et al., 2025). We consider w.l.o.g. the case $\beta = \beta_0$. Let us consider the proximal

$$p_a = \underset{h \in \mathbb{R}^{L_{\text{in}}}}{\operatorname{arginf}} \left\{ \sum_{a=1}^{L_{\text{out}}} (\tilde{\sigma}_{\beta_0}(z_0)_a - \tilde{\sigma}_\beta(h)_a)^2 \right\}, \tag{60}$$

where we recall that $\tilde{\sigma}_\beta(A) = \sigma_\beta(\sqrt{1+\mathbb{I}}\,A)$ for any matrix $A \in \mathbb{R}^{T \times T}$. It is easy to see that if $\sigma_\beta(x^*) = y$ for a symmetric $T \times T$ matrix $x^*$ and softmax with temperature $\beta$, then

$$x^*_{ab} = x^*_{TT} + \frac{1}{\beta} \begin{cases} y_{ab}/y_{aT} + y_{Ta}/y_{TT} & \text{if} \quad 1 \le a, b \le T-1 \\ y_{Ta}/y_{TT} & \text{if} \quad 1 \le a \le T-1 \quad \text{and} \quad b = T \\ y_{Tb}/y_{TT} & \text{if} \quad 1 \le b \le T-1 \quad \text{and} \quad a = T \end{cases} \tag{61}$$

This implies that (from here on we recall that $p \in \mathbb{R}^{L_{\text{in}}}$ can be seen as $T \times T$ symmetric matrix)

$$p_{ab} = z_{0,ab} + \frac{\bar{a}}{\sqrt{1+\delta_{ab}}} \tag{62}$$

where $\bar{a}$ is a scalar. Thus, we have a continuum of global minima, and we need to pick the one closest in L2 distance to $z$, giving

$$p_{ab} = z_{0,ab} - \frac{2}{T^2} \sum_{c \le d} \frac{z_{0,cd} - z_{cd}}{\sqrt{(1+\delta_{cd})(1+\delta_{ab})}} \tag{63}$$

which provides striking simplifications in the equations, that reduce to

$$\begin{cases} q & = \frac{\hat{m}^2}{\hat{\Sigma}^2} J\left(\frac{\sqrt{\hat{q}}}{\hat{m}}, \frac{2}{\hat{m}}\right) \\ m & = \frac{\hat{m} J\left(\frac{\sqrt{\hat{q}}}{\hat{m}}, \frac{2}{\hat{m}}\right) - \tilde{\lambda} \partial_2 J\left(\frac{\sqrt{\hat{q}}}{\hat{m}}, \frac{2}{\hat{m}}\right) - \frac{\sqrt{\hat{q}}}{2} \partial_1 J\left(\frac{\sqrt{\hat{q}}}{\hat{m}}, \frac{2}{\hat{m}}\right)}{\hat{\Sigma}} \\ \Sigma\hat{\Sigma} & = \frac{\hat{m}}{2\sqrt{\hat{q}}} \partial_1 J\left(\frac{\sqrt{\hat{q}}}{\hat{m}}, \frac{2}{\hat{m}}\right) \\ \Sigma\hat{\Sigma} & = 2\alpha(L_{\text{in}} - 1) \\ \hat{m}\Sigma & = 2\alpha(L_{\text{in}} - 1) \\ \hat{q}\Sigma^2 & = 2\alpha(L_{\text{in}} - 1)(Q_0 - 2m + q) \end{cases} \tag{64}$$

It is easy to see that instead, in the linear case, $p = z_0$, leading to the set of equations

$$\begin{cases} q & = \frac{\hat{m}^2}{\hat{\Sigma}^2} J\left(\frac{\sqrt{\hat{q}}}{\hat{m}}, \frac{2}{\hat{m}}\right) \\ m & = \frac{\hat{m} J\left(\frac{\sqrt{\hat{q}}}{\hat{m}}, \frac{2}{\hat{m}}\right) - \tilde{\lambda} \partial_2 J\left(\frac{\sqrt{\hat{q}}}{\hat{m}}, \frac{2}{\hat{m}}\right) - \frac{\sqrt{\hat{q}}}{2} \partial_1 J\left(\frac{\sqrt{\hat{q}}}{\hat{m}}, \frac{2}{\hat{m}}\right)}{\hat{\Sigma}} \\ \Sigma\hat{\Sigma} & = \frac{\hat{m}}{2\sqrt{\hat{q}}} \partial_1 J\left(\frac{\sqrt{\hat{q}}}{\hat{m}}, \frac{2}{\hat{m}}\right) \\ \Sigma\hat{\Sigma} & = 2\alpha L_{\text{in}} \\ \hat{m}\Sigma & = 2\alpha L_{\text{in}} \\ \hat{q}\Sigma^2 & = 2\alpha L_{\text{in}}(Q_0 - 2m + q)\,. \end{cases} \tag{65}$$

We thus see that, calling $\alpha_{\text{soft}} = (L_{\text{in}} - 1)\alpha$ and $\alpha_{\text{linear}} = L_{\text{in}}\alpha$, one get the same equations as the single-token case (Erba et al., 2025).

This implies immediately Corollary 1, giving in particular the expression for the interpolation threshold up to which the mapping discussed in this Appendix holds. It also implies immediately that the low-rank limit $\kappa_0 \to 0$ reduces to (Erba et al., 2025, Result 2) (modulo a rescaling).

# D NON-FACTORIZED ATTENTION

The case of non-factorized attention falls in the formalism of Appendix B.1, with state evolution equations given by

$$
q = \frac{\hat{m}^2}{(\hat{\Sigma} + 4\tau)^2}\left(Q_0 + \frac{\hat{q}}{\hat{m}^2}\right)
$$
$$
m = \frac{Q_0\hat{m}}{\hat{\Sigma} + 4\tau}
$$
$$
\Sigma = \frac{1}{\hat{\Sigma} + 4\tau}
$$
$$
\Sigma\hat{\Sigma} = 2\alpha L - 2\alpha\mathbb{E}_{z_0,z}\sum_{a=1}^{L_{\text{in}}}\frac{Q_0 z_a p_a - m z_{0,a} p_a}{Q_0 q - m^2}
$$
$$
\Sigma\hat{m} = 2\alpha\mathbb{E}_{z_0,z}\sum_{a=1}^{L_{\text{in}}}\frac{q z_{0,a} p_a - m z_a p_a}{Q_0 q - m^2}
$$
$$
\hat{q}\Sigma^2 = 2\alpha\mathbb{E}_{z_0,z}\sum_{a=1}^{L_{\text{in}}}(p_a - z_a)^2
$$

(66)

with $\tau$ being the Frobenius regularization. The equations can be derived by noticing that

$$
\begin{aligned}
\Psi(\hat{\Sigma}, \hat{q}, \hat{m}) &= \frac{2}{d}\mathbb{E}_D \min_{T\in\mathcal{C}}\left[\frac{1}{d^2}R(T) + \frac{\hat{\Sigma}}{4}\sum_{i=1}^{d}T_i^2 - \frac{1}{2}\sum_{i=1}^{d}T_i D_i\right] \\
&= \frac{1}{d}\mathbb{E}_D \min_{T\in\mathcal{C}}\left[\frac{\hat{\Sigma} + 4\tau}{2}\sum_{i=1}^{d}T_i^2 - \sum_{i=1}^{d}T_i D_i\right] \\
&= -\frac{1}{2(\hat{\Sigma} + 4\tau)}\frac{1}{d}\mathbb{E}_D\sum_{i=1}^{d}T_i^2 \\
&= -\frac{Q_0\hat{m}^2 + \hat{q}}{2(\hat{\Sigma} + 4\tau)}
\end{aligned}
$$

(67)

while the loss-related equations are the same as in the case of the main text.

In the limit $\tau \to 0^+$, the last three equations reduce as in the case of Appendix C, giving a set of equations that can be solved and gives (in the noiseless case)

$$
||\hat{S} - S_0||_F^2 = Q_0 - 2m + q = Q_0(1 - 2\alpha)
$$

(68)

giving the strong recovery at $\alpha = 1/2$ and proving the claim that if $\kappa_0 \to 0^+$ learning for the non-factorized model happens only at scale $\alpha = \mathcal{O}(1)$, i.e. $n = \mathcal{O}(d^2)$.

# E DETAILS OF THE IMPLEMENTATION AND ADDITIONAL EXPERIMENTS

All of the code for reproducing the figures is included in the submission and will be released upon acceptance.

The analytical predictions in all plots are obtained by iterating the equations in equation 43 until convergence. The minimization in equation 11 is performed with the minimize package of Scipy, initializing in a Gaussian of variance $10^{-4}$ centered on $z_0$. The expectations in equation 43 are computed using Monte-Carlo integration with at least $10^4$ samples. Even though a single iteration of equation 43 takes typically less than one hour on a standard laptop, for convenience we used 60 nodes with 2 Intel Xeon 8360Y CPUs. For producing this paper we used approximately 60000 CPU hours including the initial exploration.

We offer some additional numerical explorations of the model in equation 1 that are complementary for the single-head tied-attention model in the main text.

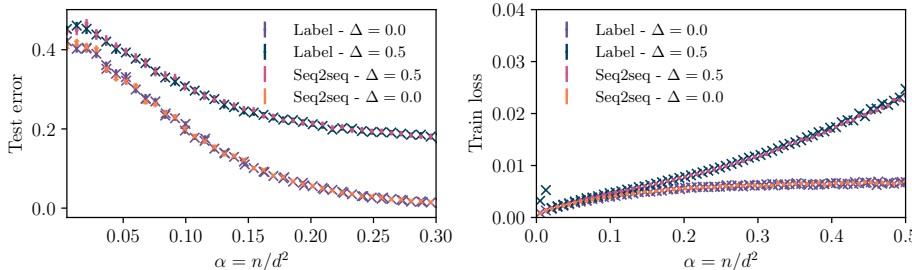

Figure 4: Comparison between the seq2seq model and the seq2lab model, both in equation 1. Both panels display the numerical simulations obtain from Adam with $\lambda = 0.01$, $\beta = \beta_0 = 1.0$, $T = 2$. We compare in both panels the noiseless $\Delta = 0.0$ and the noisy version of the model with $\Delta = 0.5$. We show in the (Left panel) the test error of the model computed from the seq2seq and the seq2lab formulas in equation 3. In the (Right panel) we show the train loss computed from the formulas in equation 2 with Adam. In all curves we average over 32 different realizations, with $d = 100$ and 2000 samples in the test set. Both variants achieve same test error and train loss, both in the noiseless and in the noisy version of the model.

We start by analyzing the asymptotic equivalence of our model of single-head tied-attention in its sequence-to-sequence formulation or in its sequence-to-label variant, both in equation 1. As a double check, we cross-check the learning curves obtained in the two variants with Adam.

In Fig. 4 we compare the sequence-to-sequence and sequence-to-label formulations, evaluating test error via equation 3 (left) and training loss via equation 2 (right), for both noiseless ($\Delta = 0$) and noisy ($\Delta = 0.5$) settings at $T = 2$. The two variants are indistinguishable within error bars across the whole range of $\alpha = n/d^2$, confirming the asymptotic equivalence used in our proof sketch. The learning curves, indeed, exhibit the same phenomenology for the two variants of the model, seq2seq and seq2lab.

We next move to analyze the impact of the inverse temperature used inside the softmax activations in both the model and the target of equation 1.

In Fig. 5 we compare the cases in which the inverse temperature matched temperatures $\beta = \beta_0 \in \{0.5, 1, 2\}$ at fixed $T = 2$. Lower temperatures (larger smoothing of the row-wise softmax) systematically yield lower test error for a given $\alpha$, both without noise (left) and with label noise $\Delta = 0.5$ (right).

To conclude, we vary the number of tokens $T$ to investigate the impact of such parameter in the model's behavior. Finally, in Fig. 6 we vary the number of tokens $T \in \{2, 3, 5\}$ and plot the test error against the rescaled sample ratio $\bar{\alpha} = n/d^2\big(T(T+1)/2 - 1\big) = 2n/(T^2 + T - 2)d^2$, motivated by the effective number of scalar constraints in the attention matrix (cf. Appendix C). After this normalization, the learning curves for different $T$ largely align in both the noiseless and noisy cases (left/right), exhibiting the same qualitative dependence on sample size and confirming that $T$ primarily rescales the usable information rather than altering the underlying learning dynamics.

## F  LARGE LANGUAGE MODELS USAGE

The authors used Large Language Models in order to polish writing.

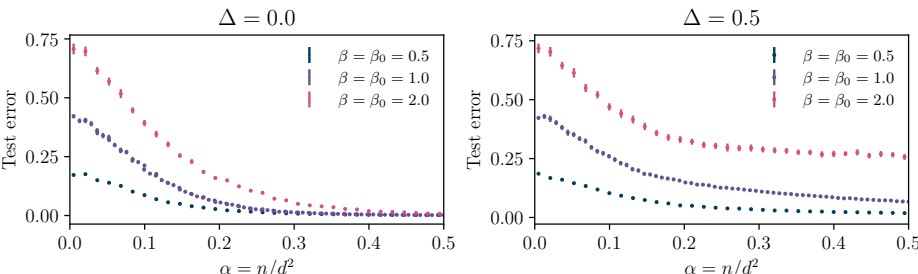

Figure 5: We show the test error for the model in equation 1 with respect to $\alpha = n/d^2$. We compute the test error through Adam for different values of the inverse temperatures $\beta = \beta_0$ of the softmax activation in the single-head tied-attention model. In all cases we consider the matched-temperature scenario between the target and the model. Both panels display the numerical simulations obtain from Adam with $\lambda = 0.01$, $\kappa_0 = 0.5$, $\kappa = 1$, $T = 2$. We compare in both panels the noiseless $\Delta = 0.0$ (Left panel) and the noisy version of the model with $\Delta = 0.5$ (Right panel) for three values of the inverse temperatures $\beta = \beta_0 = 0.5, 1, 2.0$. In all curves we average over 32 different realizations, with $d = 100$ and 2000 samples in the test set.

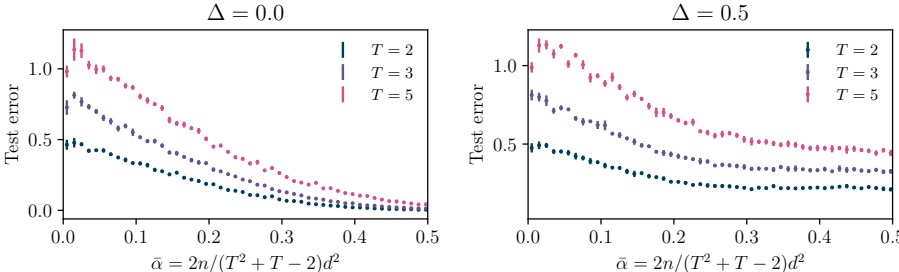

Figure 6: We show the test error computed through Adam for the model in equation 1 for a different number of tokens $T \geq 2$. In the x-axis we show the rescaled sample complexity $\bar{\alpha} = n/d^2(\frac{T(T+1)}{2} - 1) = 2n/(T^2 + T - 2)d^2$. Both panels display the numerical simulations obtain from Adam with $\lambda = 0.001$, $\kappa_0 = 0.5$, $\kappa = 1$, $\beta = \beta_0 = 1.0$ and different number of tokens $T = 2, 3, 5$. We compare in both panels the noiseless $\Delta = 0.0$ (Left panel) and the noisy version of the model with $\Delta = 0.5$. In all curves we average over 32 different realizations, with $d = 100$ and 2000 samples in the test set.

