# OpenReview forum: "Inductive Bias and Spectral Properties of Single-Head Attention in High Dimensions"
_ICLR.cc/2026/Conference — Submitted to ICLR 2026_

### Official Review · Reviewer_bdfj · 2025-10-30

**Soundness:** 3
**Presentation:** 2
**Contribution:** 2
**Rating:** 4
**Confidence:** 2

**Summary:**

This work studies the fundamental inductive bias topic of the (single-head) attention mechanism when applied to attention-indexed targets. Authors investigate this specific learning problem, derive exact formulas for optimal solutions of corresponding risk minimization, and quantify spectral laws of learned weights, under certain high-dimensional statistical asymptotics.

**Strengths:**

1. This work provides fine-grained mathematical analysis of a concrete learning problem of approximating attention-indexed targets with single-head attention.
2. The theoretical part is quite detailed, with interesting and insightful characterizations on optimal learning errors and model parameters‘ (spectrum) distributions.
3. There are numerical verifications aligned with derived theoretical results and implications.

**Weaknesses:**

1. For model space: In Eq. (1), authors only focus on a special architectural form, where for weight parameters, key and query matrices are equal (tied), and value matrices are set as identities. In addition, a centering operation and an extra $d$-scaling are required, which is not the case in practice.
2. For target space: As defined in Eq. (4, 5), the attention-indexed target also appears a special form that resembles the model. This is acceptable for approximation purposes in theory, but how does this particular form relate to practical applications?
- More questions on Eq. (5):
    - Why do we need a special scaling in input dimensions ($\sqrt{d}$) differing from that of the model ($d$)?
    - Why can we model higher-order dependencies in target input-output relationships as (Gaussian) noises?
    - It is not clear of the definition of $\Delta$ and square roots of matrices.
3. For input space: Only Gaussian input sequences are investigated. Even if this is acceptable in theory, it would be better to include numerical verifications on whether similar patterns hold in general for more types of data distributions, particularly on real-world datasets.
4. Although it is reasonable to assume the limit conditions in Eq. (6) for the purpose of exact formulas of e.g. ERM asymptotics, I wonder why such conditions are necessary. Since increasing either the model dimension $m$ or number of samples $n$ should be beneficial (particularly for the convergence of ERM), one should not restrict both of them as in Eq. (6) ($O(d)$ and $O(d^2)$, respectively).
5. More questions on Claim 1:
- It seems that there are no formal proofs of Claim 1, except the sketch of logical steps.
- The presentation is too long and can be improved for better readability. For example, authors can hide all intermediate quantities and directly state ultimate results, or use an informal version in the main text and leave the rigorous long version in appendices.
6. Again, this work can be further strengthened by both stressing theoretical insights and discussing practical implications. How can we apply the derived theory here to guide applications?

**Questions:**

See weaknesses.

---

> ### Author Response · Authors · 2025-11-21
>
> 1. *Weakness 1*:
>
> We agree that our architectural choice is a simplification of practical transformers. This is deliberate: our goal is to obtain a first exact high-dimensional asymptotic characterization of an attention layer under an attention-indexed teacher, which already requires substantial technical machinery.
>
> Regarding the analysis of a single-layer tied-attention model with value matrix set to the identity: we refer the reviewer to the answer given to Weakness 1 of Reviewer LLsC.
>
> The centering operation is analogous in spirit to a mean-only batch normalization and is technically convenient for the random matrix analysis. The $1/\sqrt{d}$ (and $1/\sqrt{md}$) scalings are standard high-dimensional scalings to keep the pre-softmax scores $(x W W^\top x^\top)$ of order one as $d,m \to \infty$, so that neither saturation nor vanishing occurs trivially. We will clarify this in a revised version of the manuscript.
>
>
>
> 2. *Weakness 2*:
>
> 2.1 : The scaling we use is set in order to have a well defined high-dimensional limit. In order to have fluctuations of order one in the attention inputs, we scale the matrix $S$ both in the target and in the architecture by $\sqrt{md}$, and we put another factor $\sqrt{d}$ due to the gaussian input data X.
> Any apparent difference of scaling is a matter of parametrization; at the level of the effective matrix problem, the teacher and student share a consistent normalization. We will make this more explicit near Eqs. (4–5).
>
>
> 2.2 : In our teacher-student setting, the target mapping is taken to be of the same functional form as the student (attention-indexed), plus an additive noise term. This is standard in high-dimensional random features and kernel learning: by construction, the student architecture is capable of representing the teacher part exactly, while the noise term captures all residual structure not modeled explicitly.
>
>
> The Gaussianity of the noise is a modeling approximation/assumption. This is motivated by central limit arguments and by analytical tractability: the variance $\Delta$ plays the role of an effective noise level / inverse signal-to-noise ratio, encoding how much of the data structure lies outside the attention-indexed component. We will clarify this interpretation and the role of $\Delta$ in the revised version.
>
> Taking the square root of matrices in this work refers to the unique positive semidefinite square root of the symmetric PSD matrix $S_0$.
>
> 3. *Weakness 3*:
>
> The point of this work is precisely the theoretical tractability and the consequent sharp results. However, we include citations (such as [1]) for the low-rank bias and spectra that are observed experimentally in factorized attention and we *quantify* sharply such phenomena in our controlled theoretical setting.
>
> [1] : Charles H Martin, Tongsu Peng, and Michael W Mahoney. "Predicting trends in the quality of state-
> of-the-art neural networks without access to training or testing data." Nature Communications, 12
> (1):4122, (2021).
>
>
>
>
> 4. *Weakness 4*:
>
> The high-dimensional scaling we focus in our work, namely that of quadratically many samples, is the critical scaling of the model we investigate. Other scalings could be considered, but result in trivial findings: for $n << d^2$ the error achieved is the one ot eh zero estimator, as the data part of the loss becomes negligible and the regularization dominates. On the other hand, for $n >> d^2$ the test error goes to zero.
>
> Regarding the rank constraint, finally, the extensive width of the target and the architecture is not necessarily required. In particular, our results apply and are independent for any width $m$ as long as $m \geq d$; we will clarify this in a revised version of the manuscript.
>
> We will emphasize in the revised text that these scalings are chosen to expose the non-trivial high-dimensional phase diagram of our model.
>
> ...
>
> The answer briefly continues in the following message

---

> > ### Author Response · Authors · 2025-11-21
> >
> > ...
> > Continuation of the previous message
> >
> > 5. *Weakness 5*:
> >
> > The reviewer is correct that we do not provide a full, self-contained proof of Claim 1, but we rather outline the steps needed.
> > The rigorous counterparts of these steps have been established for related (but simpler) models of generalized linear/quadratic networks. We rely on these works for the technical backbone and extend them to the attention-indexed setting. In particular this proof follows from standard manipulations, merging two rigorous results which are detailed in [5] and in [6], as specified in the article. In this work, we extend the setting to include multiple tokens and non-linear activations (softmax), yet this does not alter in any way the structure of the proof, as we outline in the manuscript.
> >
> >
> > [5]: Vittorio Erba, Emanuele Troiani, Lenka Zdeborová, and Florent Krzakala "The nuclear route: Sharp asymptotics of erm in overparameterized quadratic networks" NeurIPS (2025)
> >
> > [6]: Raphael Berthier, Andrea Montanari, Phan-Minh Nguyen "State evolution for approximate message passing with non-separable functions" Information and Inference (2020)
> >
> >
> >
> > 6. *Weakness 6*:
> > Our work is primarily theoretical in nature. Its main purpose is to provide a controlled setting where the inductive biases of factorized single-head attention can be analyzed exactly and related to spectral properties (low effective rank, bulk–outlier structure). These phenomena are consistently observed in experiments (see, e.g. [1]), and our results are meant to shed light and quantify such empirical obserations in an analytically controllable setting, rather than to offer direct architectural prescriptions.
> >
> > [1] : Charles H Martin, Tongsu Peng, and Michael W Mahoney. "Predicting trends in the quality of state-
> > of-the-art neural networks without access to training or testing data." Nature Communications, 12
> > (1):4122, (2021).

---

### Official Review · Reviewer_LLsC · 2025-10-31

**Soundness:** 3
**Presentation:** 2
**Contribution:** 2
**Rating:** 4
**Confidence:** 2

**Summary:**

The authors analyze ERM for single-head tied attention under both factorized (tied query and key matrices) and unfactorized settings (their product), deriving sharp asymptotics for training and test errors, interpolation thresholds, and the spectral distribution of learned weights, with emphasis on the inductive bias from weight decay.

**Strengths:**

The results are strong. The authors derive sharp asymptotics for test and training error of the global minimizer (depending on sample ratio $\alpha$ and rank ratios $\kappa$, $\kappa_0$) with experimental verification. The theoretical predictions align well with experimental results across various settings.

**Weaknesses:**

1. The neural network architecture is simplified (one-layer, tied query and key weights, no value matrix), which limits direct applicability to practical transformers.
2. Claim 1 is difficult to fully understand for reviewers outside the field. The components ($\mathcal{M}$, $J$ etc.) lack intuitive interpretation, making the main result somewhat opaque.
3. The paper analyzes only the global minimum of ERM, not the optimization dynamics of gradient methods or whether they converge to the global optimum (though experiments suggest they do).

**Questions:**

1. Can the authors explain why tied attention was chosen instead of the original untied formulation? What are the main technical difficulties in analyzing untied attention?
2. Can the authors provide a brief intuitive interpretation of Claim 1? What do the order parameters M and J represent, and what is their role in the optimization objective?
3. What is the main practical takeaway from this work? Does the analysis provide guidance for practitioners, such as how to choose the weight decay parameter $\lambda$?

---

> ### Author Response · Authors · 2025-11-21
>
> 1. *Weakness 1*:
>
> We fully agree that our architecture is a simplification of practical transformers. Our goal is to obtain a first exact high-dimensional characterization of the spectrum and generalization of an attention layer under a matching architecture, and this already requires substantial technical machinery.
> Furthermore, in the setting of one tied-attention layer, we uncover a clear phenomenology which we can entirely quantify regarding low-rank bias of factorized attention and the spectra obtained after training, mirroring experimental findings. While the full behavior of a transformer implies many interactions of different elements, here we surgically extract and anlyize what is though of as being the core, defining feature of transformers.
>
> We additionally remark that extending our full ERM/spectral analysis to untied, multi-layer or multi-head models is technically possible in principle, but leads to significantly more complex state-evolution equations and numerical challenges.
>
>
> We chose to first solve the simplest setting that still exhibits non-trivial feature learning and low-rank bias.
>
> Finally, regarding the value matrix, the focus of this work is on the attention kernel and its spectrum: values and MLPs fall outside from the pure pairwise analysis of the correlations between tokens, and are left for future explorations.
>
>
>
> 2. *Weakness 2*:
>
> The quantities introduced in the theoretical claims of this work are technical and intended for a specialized audience.
>
> In the answer to Question 2, we provide an intuitive explanation of Claim 1 and of the mathematical objects M and J, and we will try to merge this intuition in the revised manuscript.
> We stress that the best intuitive interpretation of the technical main claim of the paper is given by the phenomenology presented in the Figures, and by the summary of the results.
>
> 3. *Weakness 3*:
>
> We agree with the reviewer that the optimization dynamics of gradient-based methods is an important and complementary question. Our work explicitly focuses on characterizing the global minimizers of ERM in the high-dimensional limit. We deliberately do not make any theoretical claims about convergence rates, basin structure,  or provable global convergence of gradient descent/Adam.
> Indeed, these aspects are the subject of active research even in simpler models (e.g., quadratic networks, such as [3] and [4]), and extending such analyses to our attention setting is a challenging open problem.
>
> [3] : GB Arous, MA Erdogdu, NM Vural, D Wu "Learning quadratic neural networks in high dimensions: SGD dynamics and scaling laws", Arxiv (2025).
>
> [4] : Simon Martin, Francis Bach, Giulio Biroli, "On the Impact of Overparameterization on the Training of a Shallow Neural Network in High Dimensions" AISTATS (2024)
>
>
> 4. *Question 1*:
>
> We redirect the reviewer to the answer regarding *Weakness 1*.
>
>
>
>
> 5. *Question 2*:
> We would not call $M$ and $J$ order parameters, as they are not quantities that would describe completely the behavior of the model. In our work the order parameters are $m,q,\Sigma,\hat{m},\hat{q},\hat{\Sigma}$. Intuitively, minimizing over a function of $M$ and $J$ is equivalent of minimizing over the empirical risk in equation (2). In particular $M$ will represent the loss and $J$ the regularization.
>
>
>
> 6. *Question 3*:
>
> The main contribution of this theoretical work is to provide exact asymptotic description in a problem of learning with a tied-attention model, and to quantify in a controlled setting the low-rank bias in such models of factorized attention. In this setting, we do analytically describe and quantify spectral observations that practitioners do observe in real experiments, e.g. see [1].
>
> Rather than providing a practical takeaway we see our work as one explaining theoretically why certain practical choices are made. For instnace why the $K$,$Q$ matrices are trained separately and why weight decay is always used in current LLMs despite the fact that part of regularization certainly comes from the optimization algorithm as discussed amply in many works.
>
>
> [1] : Charles H Martin, Tongsu Peng, and Michael W Mahoney. "Predicting trends in the quality of state-
> of-the-art neural networks without access to training or testing data." Nature Communications, 12
> (1):4122, (2021).

---

### Official Review · Reviewer_gcWx · 2025-11-01

**Soundness:** 2
**Presentation:** 1
**Contribution:** 3
**Rating:** 4
**Confidence:** 2

**Summary:**

The paper studies empirical risk minimization ERM in a single-head tied-attention layer in the high-dimensional regime, trained on synthetic sequence tasks modeled using the recently proposed attention-indexed model. Leveraging random matrix theory tools, spin-glass methods, and approximate message passing technique, the authors derive sharp asymptotics for training and test errors, and thresholds for interpolation and perfect recovery. They show how weight decay induces an implicit nuclear-norm regularization promoting low-rank structure, and characterize the spectral properties of the learned weights. They compare standard factorized key-query vs. merged parameterizations, showing how the former generalizes better.

**Strengths:**

The paper is highly theoretical and adds to the line of work on training dynamics of attention models. The main strength is extending the line of work on Gaussian multi-index models to study neural networks to better understand attention models, by leveraging the recently developed attention index models. This allows considering both classification and next-token prediction under similar frameworks.

**Weaknesses:**

### 1. Missing references to several related works:

A major weakness is that the paper is missing references to some highly relevant works [1-9] on optimization dynamics and inductive bias of (self-)attention-based models (except [6] which considers fixed attention weights), for classification and next-token prediction settings. The authors should add these and contextualize their contributions taking them into account.

Specifically, the connection of GD optimization of single-head attention with implicit nuclear-norm regularization is not novel; [2] shows this connection when optimizing the key-query matrices separately with a fixed linear decoder, which explains the low-rank bias seen in practice (see section 2.1 in [2]). These works [1,2,4,5] also explain practical observations about trained attention maps being sparse due to softmax saturation.

Additionally, while the question of gradient-based methods to reach global minima for attention optimization for the setting of the paper is new, as discussed in line 378, it should be contextualized with respect to other works, e.g., [5], which study global convergence for attention optimization in a different setting which is still non-convex.

Additionally, there are also several works that study benign overfitting and training dynamics of in-context learning (ICL) in attention-ba models, which I encourage the authors to check and appropriately cite and discuss, perhaps in a related work section in the appendix. In particular, note [10], which studies ICL of linear regression in a simplified multi-head linear self-attention model, comparing the dynamics of optimizing merged key-query matrices versus optimizing them separately.

### 2. The proof of the main claim seems incomplete.

Condition 10 in the paper is not shown theoretically, and only verified empirically, as stated in lines 333-337. Can the authors clarify what the technical difficulty to prove this is, and whether any prior works have made similar simplifications?

**References:**

[1] Tarzanagh et al., “Max-Margin Token Selection in Attention Mechanism”, NeurIPS 2023.

[2] Tarzanagh et al., “Transformers as Support Vector Machines”, arxiv 2023.

[3] Tian et al., “Scan and snap: Understanding training dynamics and token composition in 1-layer transformer”, NeurIPS 2023.

[4] Tian et al., “Joma: Demystifying multilayer transformers via joint dynamics of mlp and attention”, ICLR 2024.

[5] Vasudeva et al., “Implicit Bias and Fast Convergence Rates for Self-attention”, TMLR 2025.

[6] Thrampoulidis, “Implicit Optimization Bias of Next-token Prediction in Linear Models”, NeurIPS 2024.

[7] Huang et al, “Non-asymptotic Convergence of Training Transformers for Next-token Prediction”, NeurIPS 2024.

[8] Sheen et al., “Implicit Regularization of Gradient Flow on One-Layer Softmax Attention”, arxiv 2024.

[9] Li et al, “On the Optimization and Generalization of Two-layer Transformers with Sign Gradient Descent”, ICLR 2025.

[10] Zhang et al., “Training Dynamics of In-Context Learning in Linear Attention”, ICML 2025.

**Questions:**

Please see the weaknesses section above. Some other minor suggestions for improvement:

1. In some instances, \citep and \citet are not used properly, e.g., line 62.

2. In some places, ERM is used directly to abbreviate empirical risk minimization, whereas sometimes, the full form is used later on.

---

> ### Author Response · Authors · 2025-11-21
>
> 1. *Weakness 1*:
>
> We appreciate the detailed list [1–10] and will add a dedicated related work subsection where we position our contribution more clearly.
>
> Works such as [2] and [8] (in the reviewer's comment) already show that, in certain classification settings, gradient descent/flow on separately parameterized key/query matrices induces an implicit nuclear-norm bias on the product $W = KQ^\top$, whereas directly optimizing a merged matrix favors a Frobenius-norm bias. This is indeed conceptually close to our observation that factorized vs. merged parameterizations correspond to nuclear vs. Frobenius regularization, and we will explicitly acknowledge this in the revised version. We stress that the main contribution of our work is that, in our minimal-yet-expressive analyzable model, we can quantify the induced gain.
>
> We point out that our main result is *not* the conceptual statement “factorizing induces a nuclear norm”, which is already present in [2, 8], but a *full high-dimensional asymptotic characterization* of the resulting ERM problem (with explicit weight decay) in an attention-indexed model. Among the various results, we indeed provide exact asymptotics and strict results for the performances achieved when learning such models, as well as the *complete asymptotic spectral law* of the learned attention weights.
> This kind of RMT/AMP-based exact solution of an attention model, under an attention-indexed teacher, does not seem to be covered in [1–9]. Those works focus primarily on optimization dynamics, max-margin structure, and sparsity of attention maps, often in classification or next-token prediction setups, whereas we solve the asymptotic ERM problem in a continuous regression setting.
>
> We will update the introduction to explicitly cite [2, 8] when discussing the nuclear-norm vs. Frobenius-norm effect, and clarify that our novelty lies in the quantitative solution and spectral characterization of the low-rank bias in such models of factorized attention, rather than in the qualitative observation itself, and importantly we also clarify the interplay between the spectral properties of our model with generalization in high-dimensions. This quantitative link does not seem as explicit in the citations provided.
>
> We agree that the question of when gradient-based methods reach global minima in attention models is important and has its optimization guarantees have been investigated in works such as [5, 7, 9] in various settings. Our contribution is complementary. As we remark in the main text, we do not prove convergence of gradient descent, yet we empirically verify that standard optimizers (Adam) reach minimizers with the same properties
> (test error, training loss, spectrum) as those predicted by our high-dimensional analysis for the global minimum. This is a surprising empirical observation.
>
> The analysis of the dynamics of algorithms in this specific setting is left for future work, as we believe it to require non-trivial technical advancements.
>
> We will revise the text to make this distinction clearer and to explicitly cite [5, 7, 9] as prior work on optimization dynamics in attention, and mention [10] as conceptually related in its comparison of merged vs separate K/Q parameterizations, while noting that our analysis applies also to non-linear attention (e.g. with softmax) thus accounting for more expressive architectures than the linear ones considered in [10].
>
> 2. *Weakness 2*:
>
> The reviewer is correct that Condition 10 is assumed, and not verified, in the present manuscript, due to numerical instabilities of the associated expression. Let us clarify the role of this condition, and argue that an independent test of the correctness of our theory comes from the match with numerical experiments.
>
>
>
> Condition 10 is referred as the replicon condition, and is a necessary but not sufficient condition to ensure that the results stated in our claims are rigorous.
>
> In [2], the authors show the existence of models in which gradient-descend provably converges to a zero temperature configuration despite the replicon condition being explicitly violated.
> We thus expect that there exists a more fundamental condition other than the replicon ensuring the correctness of our result, but such condition seems out of reach at the moment.
>
> An independent cross-check of our theory comes by checking numerical simulations, irrespectively of Condition 10 (which is surely true for vanishing regularization), which we present, and where we observe a perfect agreement.
>
>
> [2]: Persia Jana Kamali, Pierfrancesco Urbani, “Dynamical mean field theory for models of confluent tissues and beyond” SciPost Phys. 15, 219 (2023)
>
>
>
> 3. *Questions 1 and 2*:
>
> We thanks the reviewer for the suggestions, we will address them in a revised version of the manuscript.

---

### Official Review · Reviewer_V8P3 · 2025-11-05

**Soundness:** 2
**Presentation:** 2
**Contribution:** 2
**Rating:** 2
**Confidence:** 4

**Summary:**

This paper study the low-rank  bias of weight matrices in attention layer, in a controlled experimental settings: a single-head tied-attention trained on synthetic high-dimensional sequence-to-sequence and sequence-to-label tasks.  Authors used the RMT-tools (Marchenko-Pastur laws) to demonstrate the  bulk-outlier dynamics, and  discovered spectral laws of learned weights (in attention layers).

The findings of this papers complements the observations made in the prior work demonstrating the natural  preference for low-dimensional learning, in the context of factorized vs non-factorized $QK$-parameterization. Overall, the theoretical analysis seems aligned with the empirical evidence on the spectral behavior of attention mechanism.

**Strengths:**

1. Studying the spectral dynamics of weight matrices in Transformer architectures, particularly the attention layers, is an interesting and emerging research area, which is crucial for understanding the impact of architectural choices on the representation dynamics.

2. This work adopts a theory-first approach, which offers a more clearer understanding of how architectural design choices affect representation learning, and complements the recent empirical studies on low-rank preference and inductive biases in Transformer-based architectures.

3. The authors have provided solid theoretical justifications for the spectral laws of learned weights and inductive biased from learned (factorized vs non-factorized) $QK$-weights.

**Weaknesses:**

1. This paper does not provide any significant conceptual insights into the representation dynamics of attention weight matrix.   Prior work has already discussed the low-dimensional preference of learned representation, and the spectral dynamics of  $QK$ weight matrices in a practical multi-layer LLMs [1,2,3].


2. The practical utility of the demonstrated inductive bias of un-factorized $QK$-projections is not clear, as almost all the Transformer architectures use factorized $QK$-projection.

3. The lack of empirical power-law relationship, which is typically expected in the tails of the learned weight spectrum, limits the completeness of the spectral analysis and their utility for introducing suitable inductive bias in the existing Transformer Architecture.


[1] Dandi et al., A random matrix theory perspective on the spectrum of learned features and asymptotic generalization
capabilities,  AISTATS 2025

[2] Bao et al., self-attention networks localize when QK-eigenspectrum concentrates, ICML 2024

[3] Jha et al., A Random Matrix Theory Perspective on the Learning Dynamics of Multi-head Latent Attention, HiLD workshop ICML 2025

**Questions:**

1. What sort of power-laws (mentioned in the L#483) can author demonstrate  form the $QK$-weight spectrum? And how does it helps practitioners and  researchers to make informed architectural choices?

2. Can the authors discuss whether a power-law relationship is expected to emerge in the spectrum of attention weight matrices, similar to the scaling behavior observed in the FFN intermediate representations reported in [1]?


3. What types of inductive-biased or learning constraints can improve the tail-behavior of attention weight matrices? For example, learning weights on some particular geometrical shape ?



[1] Jha et al., Spectral Scaling Laws in Language Models: How Effectively Do Feed-Forward Networks Use Their Latent Space?, EMNLP 2025.

---

> ### Author Response · Authors · 2025-11-21
>
> 1. *Weakness 1*:
>
> We agree that *the existence* of a low-rank bias is well documented in the literature. Our contribution is not to claim its existence but to *quantitatively characterize it* in the specific context of attention, under a mathematically controlled high-dimensional setting.
> Our results provide a closed-form asymptotic characterization of the rank of the learned matrices, a precise quantification of how the rank evolves with sample complexity and regularization and analytical formulas for the optimal learned spectrum of the query/key operator (Claim 1).
>
> This is different in scope from the three works cited by the reviewer:
> - Dandi et al. (AISTATS 2025) use RMT to study the spectrum of learned features in two-layer fully-connected networks and derive sharp generalization formulas, but not for attention nor for attention-indexed teachers.
> - Bao et al. (ICML 2024) relate localization of self-attention to the variance of the QK eigenspectrum. They mainly perform an empirical spectral analysis, comparing observed spectra to classical random-matrix benchmarks (such as the Marchenko–Pastur law), but do not solve the underlying ERM problem or provide exact learning curves.
> - Jha et al. (HiLD 2025) analyze multi-head latent attention during pretraining by tracking how the eigenvalues of attention-related matrices evolve; again, the focus is empirical spectral behavior (e.g., the emergence of spikes/outliers), rather than a closed-form asymptotic solution of an ERM problem.
>
>
> To our knowledge, such a sharp, analytic description of the learned attention spectrum-derived directly from an attention mechanism (rather than from linearized models) is new. Existing empirical papers report the phenomenon qualitatively; we provide the exact asymptotic law governing it, in the simplest model we could think of, which still retains feature learning.
>
> 2. *Weakness 2*:
>
> We agree that practical Transformers use factorized Q/K projections. Our comparison with the unfactorized parameterization is not meant to suggest the latter is used in practice, but to provide a clean theoretical baseline on that fact that training $W$ in the factorized model with weight decay induces a nuclear-norm bias on the effective kernel $S = W^\top W$, and compare its learning performances with those achieved by training an unfactorized $S$ with Frobenius regularization.
>
> Our framework thus *quantifies* why the factorized parameterization-despite being less expressive-can generalize better in low-rank regimes, which helps explain a ubiquitous design choice in real architectures.
>
>
> 3. *Weakness 3*:
>
> Our theory accurately captures the full spectral phenomenology compatible with the Marchenko–Pastur (MP) data model we analyze, including emergence of a bulk-outlier transition, soft-thresholding due to weight decay and the splitting into signal and noise bulks.
> Within this model class, power-law tails are not expected:
> the learned spectrum of the attention kernel inherits the regularity of the teacher spectrum, and a MP teacher does not produce power law tails.
>
> The absence of power-law tails in our results is therefore not a limitation, but rather a reflection of the assumed generative model. This is consistent with prior empirical studies showing that power laws appear only when the data covariance or target operator themselves exhibit heavy-tailed structure. Empirical evidence of this statement is given in [1]: in the present work, we do account for all the phenomenology presented empirically apart from power-laws.
>
> In the paper we explicitly conjecture that power-law spectra would require a teacher with heavy-tailed eigenvalue distribution. This is an interesting direction, but outside the scope of the present work.
>
>
> [1] : Charles H Martin, Tongsu Peng, and Michael W Mahoney. "Predicting trends in the quality of state-
> of-the-art neural networks without access to training or testing data." Nature Communications, 12
> (1):4122, (2021).
>
>
> ...
> The answers to the questions continues in the following message

---

> > ### Author Response · Authors · 2025-11-21
> >
> > ...
> >
> > Continuation of the previous message.
> >
> >
> > 4. *Question 1*:
> >
> > Under the MP teacher considered in our theorem no power law is expected, and our analysis correctly reflects this. The spectral law we derive is the exact asymptotic solution of the nuclear-norm regularized matrix estimation problem corresponding to attention.
> >
> >
> > Our goal is not to prescribe specific architectural choices, but to provide a precise limiting description of what factorized attention with weight decay learns in a solvable setting. This description already helps interpret several empirical observations (bulk–outlier split, effective low rank, benign width overparameterization), even though it does not produce power-law tails under the MP assumption.
> >
> >
> > 5. *Question 2*:
> >
> > The spectral power law behavior reported in the cited work for Feed-Forwards Networks (FFN) [1] concerns how the spectral ranks of FFN weight matrices scale with width across large, deep language models (LLaMA, GPT-2, etc.) trained on natural data. The authors observe that a “soft” spectral rank follows an almost perfect power law in the FFN hidden dimension, while a “hard” rank grows more slowly, reflecting how deep, residual, multi-layer FFN stacks use their latent space.
> >
> > Our setting is very different: our model isolates a single attention head with Gaussian isotropic inputs, fixed sequence length and a Marchenko-Pastur spectrum in the target, thus lacking the architectural ingredients that induce scale-free behavior in the aforementioned work. Within our assumptions, power-law tails or scaling are therefore not expected.
> >
> > That said, our framework suggests how such behavior could arise in principle: if one replaced the Marchenko–Pastur teacher by a heavy-tailed spectral measure, or embedded our block into a deeper, residual architecture, the learned spectrum would inherit more complex tails, and power-law–like behavior could emerge. Analyzing these extensions is a natural direction for future work.
> >
> > 5. *Question 3*:
> >
> > The tail behavior in our model is controlled by the teacher spectrum (data model) and by the regularizer induced by the parameterization. Factorized W with weight decay leads to a nuclear-norm shrinkage (soft-thresholded spectrum), while considering unfactorized S leads to a ridge-type spectrum.
> >
> > Changing the tail would require changing the data model or the regularization, for example using a heavy-tailed target (e.g., power-law covariance) or introducing explicit sparsity or spectral penalties.
> > These extensions are beyond the scope of our current analysis, but our framework can, in principle, accommodate them.

---

### Author Response · Authors · 2025-11-27

We thank all reviewers for the reviews received.

We have updated the manuscript accordingly. The main changes are:

- In lines 042-044 and lines 079-082: we added and discussed the references [1,2] suggested in the reviews on implicit regularization and low-rank bias in attention networks, including works on implicit nuclear-norm vs Frobenius-norm bias and explicit $\ell_2$ regularization in attention.  We also clarified how our factorized vs. unfactorized comparison relates to [3] provided in the reviews.
- In lines 107-111 and lines 154-155: we clarified the role of the scalings $\sqrt{dm}$ and $\sqrt{d}$ in the definition of the attention scores.  We further explained the interpretation of the noise parameter $\Delta$ as an inverse signal-to-noise ratio.
- In lines 168-172: we expanded the discussion around the scaling $\alpha = n/d^2$, explaining why it is the critical regime: for $\alpha \to 0$ the ERM reduces to the null estimator (pure regularization), while for $\alpha \to +\infty$ perfect recovery is achieved.  We also clarified that our results do not require $m$ to scale linearly with $d$, but apply for any $m \ge d$.
- In lines 186-188: we emphasized that our results connect the global landscape properties of the learned attention weights (their spectrum and effective rank) to generalization in a controlled teacher–student setting, formalizing spectral phenomena observed empirically in large models.
- In lines 413-418: we added a short discussion situating our work with respect to recent results on optimization guarantees for attention, in particular the works [4,5,6] cited in the reviews.  We clarified that we do not provide convergence guarantees for gradient-based methods in our setting, and we explicitly leave the analysis of the algorithmic dynamics in our high-dimensional regime as an interesting direction for future work.

[1] : Tarzanagh et al., “Transformers as Support Vector Machines”, arxiv 2023.

[2] : Sheen et al., “Implicit Regularization of Gradient Flow on One-Layer Softmax Attention”, arxiv 2024.

[3] : Zhang et al., “Training Dynamics of In-Context Learning in Linear Attention”, ICML 2025.

[4] : Vasudeva et al., “Implicit Bias and Fast Convergence Rates for Self-attention”, TMLR 2025.

[5] : Huang et al, “Non-asymptotic Convergence of Training Transformers for Next-token Prediction”, NeurIPS 2024.

[6] : Li et al, “On the Optimization and Generalization of Two-layer Transformers with Sign Gradient Descent”, ICLR 2025.

---

### Meta-Review · Area_Chair_2FFG · 2026-01-06

**Summary:**

1. Novelty / positioning vs. prior work unclear; missing key related work
2. Practical relevance unclear / limited applicability to real Transformers
3. Model/assumption simplifications: tied attention, no value matrix, centering and special scalings
4. No power-law tails in spectra; analysis “incomplete” without tail scaling laws
5. Focus on global ERM optimum only; no theory of optimization dynamics
6. Rigor gaps, unclear technical assumptions: incomplete proof steps and stability condition
7. Presentation/readability issues (notation, length, clarity of main results, minor writing issues)

**Reviewer Concerns:**

The reviewer concerns about the clarifications on the technical assumptions and claims are largely addressed. The main concerns that are still outstanding are about the simplifying assumptions (e.g., tied attention weights and no value weights) and the limited relevance to real transformers used in practice, in which case the implications of the developed theory are a bit vague.

**Reviewer Scores:**

1. Reviewer V8P3 would increase their score to 4 or 5, as their questions regarding power-law behaviors have been addressed, but the practical utility of the developed theory remains unclear.
2. Reviewer gcWx would increase their score to 5 or 6, as their comments about missing references and other clarification questions should have been addressed by the authors' rebuttal.
3. Reviewer LLsC might slightly increase their score to 5, but their concerns about the practical takeaway remains.
4. Similar to the previous reviewer, Reviewer bdfj might slightly increase their score to 5.

---

### Decision · Program_Chairs · 2026-01-26

Reject